# A Mechanistic Study of Transformers Training Dynamics

## Abstract

Large-scale pretraining of transformers has been central to the success of foundation models. However, the scale of those models limits our understanding of the mechanisms at play during optimization. In this work, we study the training dynamics of transformers in a controlled and interpretable setting. On the sparse modular addition task, we demonstrate that specialized attention circuits, called "*clustering heads*", can be implemented during gradient descent to solve the problem. Our experiments show that such pathways naturally emerge during training. By monitoring the evolution of tokens via a visual sandbox, we uncover a two-stage learning and the occurrences of loss spikes due to the high curvature of normalization layers. Our findings provide several insights into patterns observed in more practical settings, such as the pretraining of large language models.

## 1 Introduction

Transformers (Vaswani et al., 2017) have become the default backbone of state-of-the-art models in a wide range of domains, including natural language processing (NLP, Brown et al., 2020; Touvron et al., 2023a), computer vision (Caron et al., 2021; Dosovitskiy et al., 2021), time series forecasting (Ilbert et al., 2024; Nie et al., 2023), and mathematical reasoning (Comanici et al., 2025; Guo et al., 2025). The models' size and amount of training data have drastically increased as a by-product of the scaling hypothesis (Charton, 2024; Hoffmann et al., 2022; Kaplan et al., 2020). This makes the analysis and ablation studies on such systems challenging. As a result, our understanding of the *training dynamics* and causes of instability in large transformer models remains limited, with much of the knowledge in this area relying on heuristic tricks and techniques.

**Mechanistic viewpoint.** To address these limitations, we draw inspiration from the field of *mechanistic interpretability*, which aims to understand the internal mechanisms at play in large language models by reverse-engineering their computations (Elhage et al., 2021). A common approach in this framework is to study the pathways, or *circuits*, of a neural network that transform an input into a given output. This viewpoint has been used to study the emerging capabilities of LLMs with in-context learning (ICL) (Edelman et al., 2024; Olsson et al., 2022), chain-of-thought (Cabannes et al., 2024a), and memorization in transformers (Bietti et al., 2023). In line with these studies, we aim to design a controlled setup to decipher the training of transformers without potential parasitic influence. Our focus is on obtaining a finer-grained understanding of the evolution of sequence embeddings during the training of a single transformer block. Training the whole architecture could then be considered as the joint training of several blocks, with the inputs of a given block being the output of its preceding block.

**Our approach.** Motivated by the fact that many NLP and reasoning tasks involve invariances in the data (Robinson & Wingate, 2023; Singh et al., 2025), we consider the *sparse modular addition* problem, which requires a correct handling of invariances to be solved and ensures non-trivial optimization dynamics (Odonnat et al., 2025). Similar mathematical tasks have been used in prior works for their clear structure and ease to observe the pathways implemented (Ameisen et al., 2025). We frame the problem as a supervised question-answering task and train small transformers on it. This enables us to fully characterize how transformers learn to solve the task, which is a step towards opening the black-box (Charton, 2024; Koh & Liang, 2017). To analyze the training dynamics on this task, we view sequences of tokens as a system of interacting particles

and design a visual sandbox to observe their evolution during training. It allows us to identify and analyze multiple phenomena observed in more practical settings. This showcases the benefits of our setup in studying, at a small scale, the training behavior of bigger models.

**Our contributions.**   Our main contributions can be summarized as follows:

1. **Theoretical analysis:** We frame the *sparse modular addition* problem as a question-answering task and demonstrate that transformers trained with gradient descent implement specific pathways, dubbed *clustering heads*, to solve it by capturing the invariances in the data.
2. **Visual sandbox:** We design a visual sandbox to observe the evolution of tokens, seen as interacting particles, during training and use it to study the implementation of clustering heads.
3. **Training dynamics:** We observe that such circuits emerge after a two-stage learning, akin to the in-context learning settings (Olsson et al., 2022). We observe the occurrence of loss spikes during training and suggest potential strategies for mitigation, relating our findings to the pretraining of large language models.

We hope our setup can serve as a viable testbed to study training stability in small scale to gain intuition for larger-scale settings.

## 2   Methodology

Throughout the paper, we use the notation $[n] = \{1, \ldots, n\}$ to denote a finite vocabulary space for $n \in \mathbb{N}$. The Euclidean norm of a vector $x \in \mathbb{R}^n$ is denoted by $\|x\|$. By abuse of notation, we denote by $\|A\|$ the entry-wise $\ell_2$-norm of a matrix $A \in \mathbb{R}^{n \times m}$ which amounts to computing the Euclidean norm of $A$ flattened. The probability simplex over $\mathbb{R}^n$ is denoted by $\Delta_n$. The total variation distance between two probability distributions $p, q$ is denoted by $d_{\text{TV}}(p, q)$.

**Sparse modular addition.**   This problem is inspired by the sparse parity problem (Abbe et al., 2023; Barak et al., 2023) and the modular addition problem (Nanda et al., 2023; Power et al., 2022) and is characterized by the following parameters:

- Input length $L \in \mathbb{N}$,
- Vocabulary size $n \in \mathbb{N}$,
- Sparsity index $k \in [L]$, and a set of indices $I \subset [L]$ with cardinality $|I| = k$.

Our default configuration is set to $L = 12$, $k = 5$, and $n \in \{2, 3\}$. Without loss of generality, we assume $I = [k] := \{1, 2, \ldots, k\}$. Inputs are sequences of $L$ tokens $x_t$ in the ring $\mathbb{F}_n = \mathbb{Z}/n\mathbb{Z} \simeq [n]$, and the corresponding targets are the sum of the first $k$ terms modulo $n$. Formally, we aim to learn a mapping:

$$
f^* : \quad \begin{array}{ccc} (\mathbb{F}_n)^L & \to & \mathbb{F}_n \\ x = (x_1, \ldots, x_L) & \mapsto & \sum_{t \in [k]} x_t. \end{array} \tag{SMA}
$$

This mapping defines a deterministic conditional distribution linking input and output data through the formula $p(y|x) = \mathbf{1}_{\{f^*(x) = y\}}$. The (SMA) problem can be framed as a supervised question-answering task with tokens embedded in a vocabulary space $\mathcal{V} = [n]$ of size $n$. Input questions are represented as sequences of tokens. They are randomly sampled and used to train sequence-to-sequence transformers by minimizing the cross-entropy loss between the model's predictions and the correct answer, that is, the sum of the $k$ first terms of the inputs modulo $n$. While sparse modular addition might seem simple to solve for a human, this is a hard task to solve for neural networks, which have been shown to struggle on similar arithmetic tasks (Charton, 2024; Dziri et al., 2023; Lee et al., 2024; Saxton et al., 2019), involving non-trivial learning dynamics (Odonnat et al., 2025). In particular, the model needs to capture the invariance in the data and correctly focus on the informative, i.e, *non-spurious*, tokens $x_i$ for $i \in [k]$.

**Transformer architecture.**   To conduct our analysis, we consider a simplified transformer model containing all the key elements of its larger counterparts commonly used in question-answering tasks (Raffel et al., 2023; Vaswani et al., 2017). Notably, it relies on a cross-attention module, a feedforward network, residual

connections, normalization layers, and weight tying (Press & Wolf, 2017b). Following modern models (Llama Team, 2024), we use a pre-norm configuration (Xiong et al., 2020) and the RMSNorm (Zhang & Sennrich, 2019), which is similar to LayerNorm (Ba et al., 2016) but computationally more efficient. To ease notations, we incorporate the RMSNorm trainable parameters into the attention and feedforward weights following Castin et al. (2024), which does not change the expressivity or optimization of the model. We recall below how tokens are processed through the model. Since the sparse modular addition problem is inherently discrete, we first embed each token in dimension $d$ with both semantic and positional information before normalizing it on the unit sphere $\mathbb{S}^d$. Formally, given a learnable token embedding $E : \mathbb{F}_n \to \mathbb{R}^d$, and a learnable position embedding $P : [L] \to \mathbb{R}^d$, a sentence in token space is lifted to a sentence in embedding space $z \in \mathbb{R}^{d \times L}$ through the following operation that applies to each $x_t$ for $t \in [L]$

$$z_t := Z(x_t, t) := \frac{E(x_t) + P(t)}{\|E(x_t) + P(t)\|}. \qquad \text{(token embedding)}$$

Then, a cross-attention mechanism is applied to the sequence of tokens $z$, leading to a single sentence embedding $\xi \in \mathbb{R}^d$. The sentence embedding can be written in matrix form as

$$\xi := (Vz)\,\mathrm{softmax}\Big(\frac{z^\top q}{\sqrt{d}}\Big) \in \mathbb{R}^d, \qquad \text{(sentence embedding)}$$

where we merge the query and value into a vector $q \in \mathbb{R}^d$ to avoid cumbersome notations to Zhang et al. (2024a). We omit the key matrix, which would act as extra parameters that do not increase the expressivity of our model since $z$ can be set to anything thanks to $E$ and $P$. The sentence embeddings are then passed through a feedforward network implemented as an MLP with two layers $U \in \mathbb{R}^{d \times h}$ and $W \in \mathbb{R}^{h \times d}$, followed by a residual connection. It implements the following transformation:

$$\psi := \xi + U\sigma\left(\frac{W\xi}{\|\xi\|}\right) \in \mathbb{R}^d \qquad \text{(sentence transform)}$$

with $\sigma \colon x \mapsto x\varphi(x)$ the GeLU activation (Hendrycks & Gimpel, 2023). $W$ receives the sentence embeddings before projecting them into a higher dimension, while $U$ reassembles the embeddings, projecting them back to the original dimension. As such, the feedforward can be equivalently seen as a combination of $h \in \mathbb{N}$ "receptors" weights $w_i \in \mathbb{R}^d$ and $h$ "assemblers" vectors $u_i \in \mathbb{R}^d$ for $i \in [h]$, with $(u_i)_{i=1}^h$ the columns of $U$ and $(w_i^\top)_{i=1}^h$ the rows of $W$. The final sentence vector $\psi$ is decoded back to token space based on how it aligns with the respective token embeddings, which amounts to using weight-tying in the implementation (Press & Wolf, 2017a). More specifically, the model's logits write

$$\zeta = (E(v)^\top \psi)_{v \in \mathbb{F}_p} \in \mathbb{R}^n. \qquad \text{(logits)}$$

Abstracting all the learnable weights into a single vector $\theta$, the probability distribution over answers in $[n]$ writes $\hat{p}_\theta(\cdot|x) \in \Delta_n$ and is obtained by applying a softmax layer on top of the logits. The model is optimized by minimizing the cross-entropy loss between its prediction and the correct answer. The loss writes

$$\mathcal{L}(\theta) := \mathbb{E}\left[-\log(\hat{p}_\theta(y|x))\right], \qquad \text{(loss)}$$

where the expectation is taken over the distribution of training samples $(x, y)$.

## 3 Learning clustering heads by gradient descent

In this section, we study how transformers learn to solve the (SMA) problem with gradient descent. We describe a natural circuit, dubbed *clustering head*, that, if correctly implemented, enables the transformer to solve the task, and we demonstrate that such a circuit can be efficiently learned by gradient descent.

**Circuit perspective.** In the sparse modular addition problem, the output $y$ is invariant to two sets of transformations of the inputs $x = (x_1, \ldots, x_L)$.

- "Permutation invariance:" $y$ does not depend on permutation of non-spurious tokens, i.e. for any permutation $\sigma$ of $\{1, \ldots, k\}$,

$$f^*(x) = f^*((x_{\sigma(1)}, \ldots, x_{\sigma(k)}, x_{k+1}, \ldots, x_L)).$$

- "Suffix invariance:" $y$ does not depend on the suffix $(x_{k+1}, \ldots, x_L)$.

These two sets of invariants can be easily enforced by the embedding layer. Any architecture where the position embeddings satisfy $p_t = p_1$ for $t \in [k]$, will be permutation invariant, meaning that its output will be invariant to permutations of the non-spurious tokens. Similarly, suffix invariance can be enforced by ensuring that the query vector primarily aligns with the token embeddings $z_t$ for $t \in [k]$, allowing the sequence embedding $\xi$ to be invariant to the sequence suffix. Such a construction would yield $\binom{k+n-1}{k}$ clusters of sequence embeddings,[1] which the feedforward layer could scatter into as many decision regions to map each sequence embedding $\xi$ to the correct output class $y \in [n]$. The pathway that realizes such operations is called a *clustering head*. Clustering heads define one way for the transformer to deal with invariances, where the model first attend to the non-spurious tokens $(x_i)_{i=1}^k$, before regrouping them by values to determine the parity of each subgroup and finally of the whole sequence. Although this is not the only possible circuit, it resonates with recent studies (Lindsey et al., 2025; Odonnat et al., 2025), which showed that transformers and notably large language models split hard problems into simpler sub-problems, for instance, to compute a complex arithmetic task.

**Learning clustering heads.** A natural question is whether *clustering heads* can be implemented in practice by a transformer trained via gradient descent, which aims at finding local minimizers of the loss $\mathcal{L}$. Such minimizers are stationary points, i.e., solutions of $\nabla \mathcal{L} = 0$. Akin to Nichani et al. (2024), we focus on the attention and feedforward dynamics, assuming fixed embedding weights. In the following lemma, we provide a closed-form expression of the gradients (the full statement is deferred to Lemma 2).

> **Lemma 1** (Closed-form gradients). *Let $m = d(1 + d + 2h)$. The trainable parameters of the transformer write $\theta \in \mathbb{R}^m$, encompassing the attention and feedforward matrices. Denoting by $\hat{p}_\theta(\cdot|x)$ and $p(\cdot|x)$ the model's prediction and ground-truth for an input sequence $x = (x_1, \ldots, x_L)$, we have*
>
> $$\nabla \mathcal{L}(\theta) = \sum_{j=1}^n \mathbb{E}_x [(\hat{p}_\theta(y = j|x) - p(y = j|x)) \cdot \nabla_\theta \zeta_j].$$

Computing the loss gradient involves reweighting each logit's gradient by the corresponding discrepancy between the model's prediction $\hat{p}_\theta$ and the ground-truth $p$. Lemma 1 allows us to characterize the stationary points of the loss and show that the parameters of transformers that implement clustering heads are stationary points (the full statement is deferred to Proposition 3).

> **Proposition 1** (Learning clustering heads, informal). *Let $\theta$ be the parameters of a transformer model that implements a clustering head. Then, $\theta$ is a stationary point of the loss, that is $\nabla \mathcal{L}(\theta) = 0$.*

Proposition 1 implies that clustering heads can be learned by gradient descent. We note that it does not provide guarantees on the convergence to such a circuit (we will see in the next section that, in practice, clustering heads are implemented by transformers). Further theoretical treatment to characterize the stationary points as local, global minima, or saddle points is an interesting direction left for future work. In the following, we focus on the training dynamics during gradient descent.

## 4 Visualizing the training dynamics

In this section, we see tokens as interacting particles, and design a visual sandbox to observe their evolution during training. It allows us to study the practical implementation of clustering heads, noticing a two-stage

---

[1]This number corresponds to the number of ways to split $k$ into $n$ buckets, which is also the number of stars and bars configurations with $k$ stars and $n - 1$ bars.

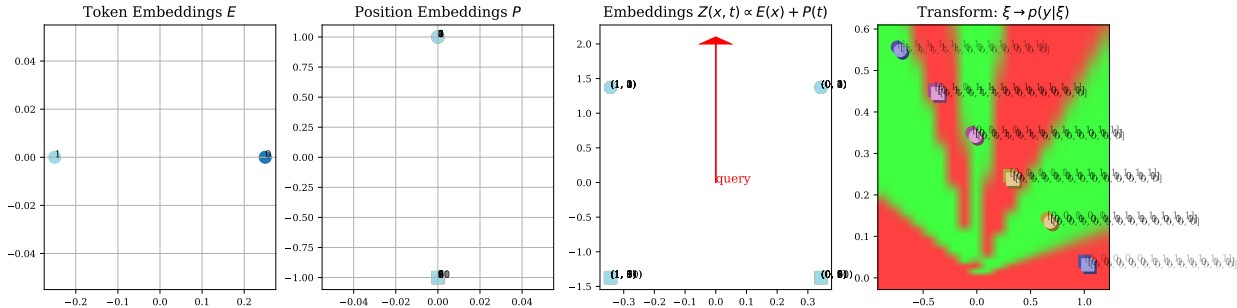

Figure 1: **Clustering Head**. Implementation of an idealized circuit that captures the invariants of the problem. *From left to right*: we first plot the two token embeddings $E(x)$ for $x \in [p]$ with $n = 2$. We then plot the twelve position embeddings $P(t)$. The spurious (resp. non-spurious) positions are represented with squares (resp. circles), and all collapse to a single point. The normalized embeddings $Z(x,t)$ are plotted in the third frame, annotated with $(x,t)$, with the query vector $q$ represented as a red arrow. This arrow points in the direction of the embedding $Z(x,t)$ for $t \in [k]$, allowing the attention mechanism to focus exclusively on non-spurious tokens. Finally, we plot the sequence embeddings $\xi$ for some $(x_t)$, and the output of the feedforward transform. The embeddings are clustered and respect the invariance, and the feedforward layer can then map each cluster to a probability vector $\hat{p}_\theta$. Each output class is represented by a color.

learning and the occurrence of loss spikes during training, and further relate our findings to the pretraining of large language models. The implementation details are given in Section A.

## 4.1 Visual sandbox

Tokens processed by transformers interact through the attention module, which allows us to study their evolution as a dynamic system of interacting particles controlled by an ordinary differential equation (Geshkovski et al., 2023b, 2025; Lu et al., 2019). We adopt this macroscopic viewpoint and follow the mechanistic viewpoint (Olah, 2023) by embedding all computations in the plan by taking $d = 2$ and tracking several key aspects during training, such as:

**Position embeddings.** We visualize them as a point cloud with the "spurious" embeddings $P(t)$ for $t \notin [k]$ represented by squares, and the "non-spurious" ones, the $P(t)$ for $t \in [k]$, by circles. Ideally, the transformer would collapse all spurious (resp. non-spurious) position embeddings into a single point, learning invariance of $y$ to sentence suffixes (resp. to permutation of non-spurious tokens positions).

**Token embeddings.** We visualize the token embeddings $Z(x,t) \propto E(x) + P(t)$ for $(x,t) \in \mathbb{F}_n \times [L]$. We maintain the same circle and square distinction and use the same color for both $E(x) + P(t)$ and $E(x') + P(t)$, for $x' \in \mathbb{F}_n \setminus \{x\}$. On the normalized plot, we also plot the query $q$ as an arrow in $\mathbb{R}^2$, helping us understand where the attention module learns to focus.

**Attention map.** We visualize the concatenation of attention vectors for different sentences. It enables us to follow the change in activation patterns, even though it is a pure function of the normed embedding visualization.

**Value transform.** We visualize $VZ(x,t)$ as a point cloud. This allows us to understand how sequence embeddings are built and how the value matrix may overcome faulty attention patterns.

**Sequence embeddings and transforms.** We visualize the sequence embeddings $\xi$ (or their transforms $\zeta$) for a set of predefined sentences. These sentences are built by iterating over prefixes $(x_t)_{t \in [k]}$ and suffixes $(x_t)_{t \notin [k]}$. Sentences that share the same prefix have the same color. Sentences whose prefixes are equivalent up to a token position permutation have similar colors. Squares, circles, and triangles are used to distinguish

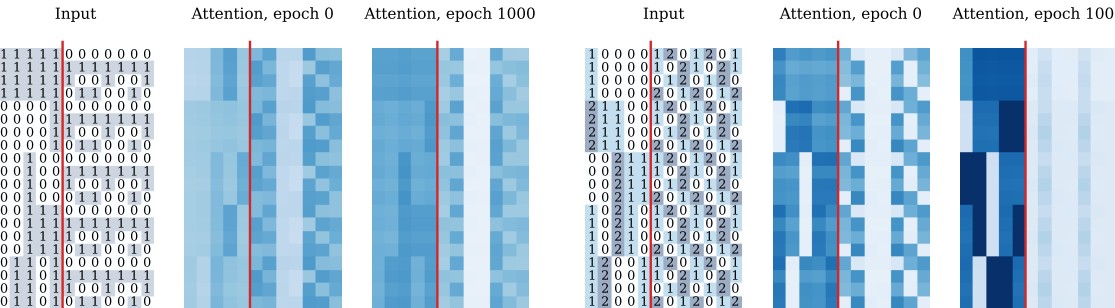

Figure 2: **Impact of the initialization.** For each input sequence (**left**), we plot the attention weights at the start of training (**middle**) and at the end of training (**right**); the darker, the higher. Final attention maps correlate with the original one, illustrating the impact of the original configuration on the implemented circuit.

Figure 3: **Transferability of circuits**. This study is akin to Fig. 2 when the model is first pre-trained with $n = 2$ and then finetuned with $n = 3$. This illustrates that the final variations of the original circuit depend on the original weight configuration. We notice that the model learns better to focus on the non-spurious first 5 tokens.

between the classes of the sentences. The sequence embeddings visualization is a direct function of the value transformation and the normalized embeddings.

**Transform level lines.** We visualize the mapping from sentence embedding $\xi$ to their associated learned probabilities $\hat{p}_\theta(y|x)$. We also plot the sentence embedding on the same plot to better understand the level line changes.

**MLP receptors and assemblers.** We visualize the $w_i \in \mathbb{R}^2$ (and $u_i \in \mathbb{R}^2$) that define the MLP transform as a point cloud in $\mathbb{R}^2$. A consistent color scheme is used to link receptors with the corresponding assemblers.

**Loss and accuracy.** We visualize the current train loss, test loss, and accuracy. It is interesting to put those classical quantities in relation to the other visualizations to better understand the loss spikes, loss plateaus, and phase transitions.

### 4.2 Clustering heads implementation

The *clustering head* described in Section 3 can be visualized in Fig. 1, with the model focusing on the informative tokens to regroup them by values with the attention module before computing the parity of the sequence with the feedforward. In the previous section, we showed that these pathways could be effectively learned by gradient descent. Since transformers can perform similar operations in many different ways, we observe in practice variations from the idealized circuit: the attention is compensated by the value matrix to capture well the invariance, the sentence embeddings present more clusters than the idealized model, the clustering is unconventional, but the model still generalizes well (see Figs. 11 to 13). We observe in Fig. 2 the strong similarity between the attention patterns at the start and at the end of the training, indicating that initialization plays a crucial role in the concrete realizations of clustering heads. We notice that the transformer learns to focus on the 1 among the non-spurious tokens, before counting them and deducing the number of 0 to make its final prediction.

**Transferability of circuits.** Among the lessons of training very large models is that some sources of data may facilitate the learning of certain skills (Cabannes et al., 2024a; Llama Team, 2024; Touvron et al., 2023b), suggesting the transferability of circuits. We illustrate this phenomenon in Fig. 3, which shows that training first a model on a simpler task (e.g., $n = 2$) helps it better solve a more challenging one (e.g., $n = 3$), as can be seen in along with the impact of initialization (more details in Section A.3).

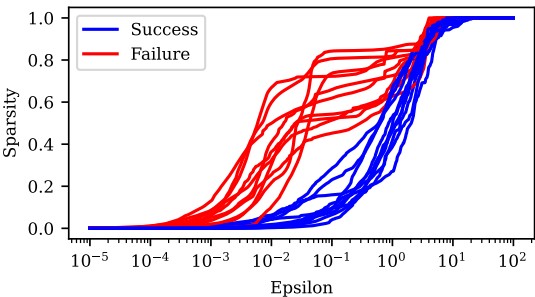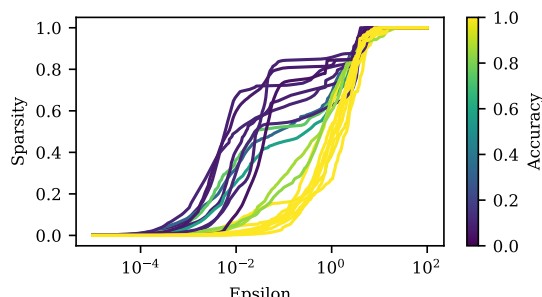

Figure 4: *Connection between performance and sparsity.* Since the activation is the GELU that does not map all the negative entries to 0 (contrary to the ReLU), we define the activation sparsity as the percentage of entries with values lower than $\varepsilon > 0$. We display the evolution of the activation sparsity of 20 trained models with $\varepsilon \in [10^{-5}, 10^2]$. **Left:** Successful models (i.e., with test accuracy above 0.9) in blue have less sparse activation than failed models in red. **Right:** The color indicates the models' test accuracy (the lighter, the better). The performance increases as the activation sparsity decreases.

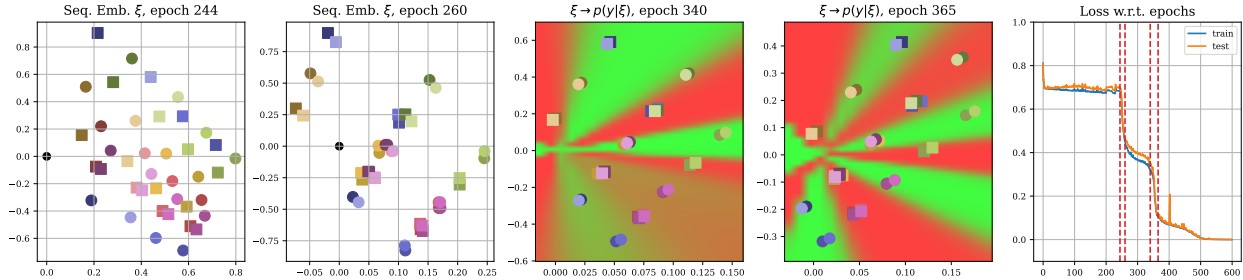

Figure 5: **Two-phase learning**. *From left to right:* **(1)** During the first snapshot, the sequence embeddings lack any clear structure. **(2)** They suddenly become clustered after the first loss drops, as seen in the second snapshot. **(3)** At this point, the MLP already classifies some clusters correctly (third snapshot). **(4)** A second loss drop occurs as the MLP gets fitted (last snapshot). *Right:* Loss profile featuring two significant drops in loss, marked by four red dashed lines at key snapshots.

**Connectivity of neurons.** Motivated by recent works studying the sparsity of transformers' activations (Li et al., 2023a; Mirzadeh et al., 2024), we investigate how neurons connect to implement *clustering heads.* We show in Fig. 4 that the models that successfully solve the task have denser activation, suggesting that many neurons are necessary to correctly implement a clustering head and thus capturing the invariance of the task (more details in Section A.2).

### 4.3 Two-stage learning: escaping saddle points

To better understand how clustering heads emerge, we observe the evolution of the tokens along with the loss landscape during training. We notice a two-stage learning process with the model first clustering the sequence embeddings, before fitting a classifier on top. As displayed in Fig. 5 (right), the loss curves present two drops, corresponding to the learning of different parts of the network. Our findings resonate with theoretical arguments in the deep learning theory literature on saddle point (Chi et al., 2019a; Dauphin et al., 2014; Du et al., 2017) and stage-wise transitions of in-context learning of transformers (Edelman et al., 2024; Hoffmann et al., 2024; Minegishi et al., 2025; Olsson et al., 2022; Reddy, 2024; Varre et al., 2025).

**First loss drop: learning of the sequence embeddings.** The first loss drop coincides with the learning of the sequence embeddings. It corresponds to a phase change in the dynamics of the weights. Before the first phase change, the weights seem to wander as if trapped in a saddle point, waiting for a clear signal to escape it. At one point, they all move quite rapidly to create a relatively definitive structure for the sequence

embeddings. Interestingly, we also notice that the time it takes for this phase change to occur when changing the training hyperparameters can vary quite a lot, reflecting the highly unpredictable time needed to escape from the saddle point. This is illustrated in Section A.4 where Fig. 16a shows the high variability of the test accuracy after 1000 epochs when slightly varying one hyperparameter (Fig. 16b shows the same phenomenon with more regularity when averaging over the runs). The exit from a saddle point can also be understood via the gradient norms, as discussed at the end of this section.

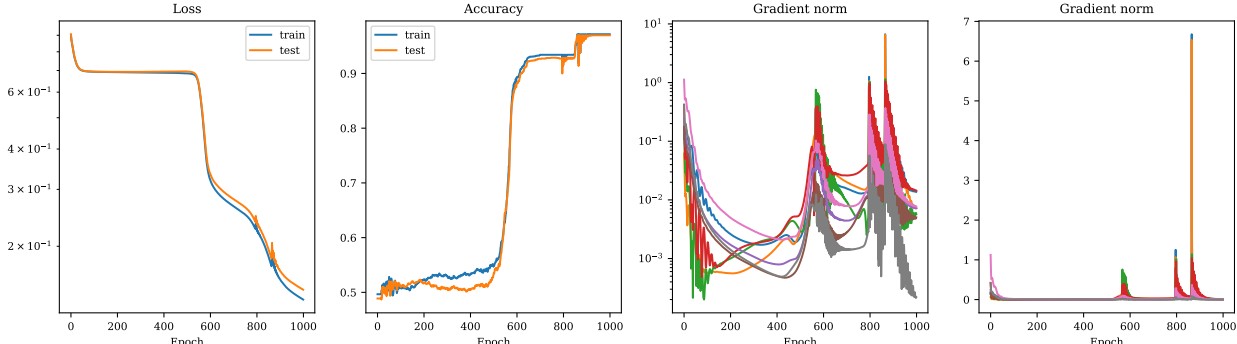

Figure 6: **Escaping saddle points**. *From left to right:* Evolution of train and test losses, the corresponding accuracies, the evolution of gradient norms for each layer in log-scale, and the similar evolution in linear scale in full-batch. We see that the learning phases of Section 4.3 appear in tandem with high gradient norms. This can be seen in the last subfigure, where the three pics correspond to the loss drops and their corresponding plateaus.

**Second loss drop: fitting of the MLP.** The second loss drop is due to the learning of the feedforward network. This change is about fitting the MLP weights to assign the correct classes to the different clusters created during the learning of the sequence embeddings. Interestingly, this second loss drop does not correspond to a clear phase change in the dynamics of the weights. The MLP weights seem to evolve at a continuous speed, although the corresponding decision frontiers change relatively strongly. We notice that this second loss drop appears soon after the first one, if not simultaneously. When the model's training stagnates, the loss plateaus for several epochs before decreasing again. This behavior may be understood by the optimizer reaching a saddle point in the loss function w.r.t. the model's parameter, that is, the gradient of the loss is (almost) zero, but the optimization has not yet reached a local minimum. The connection between gradient norms and learning phases is salient in Fig. 6, where we can see that loss drops occur in tandem with high gradient norms for each layer.

**Theoretical insights.** Proposition 2 characterize how the gradient magnitude is controlled along the training (the full statement is deferred to Proposition 4).

**Proposition 2** (Gradient upper-bound, informal)**.** *Let $\theta$ be the transformer's parameters. Assuming bounded token embeddings, the gradient of the loss verifies*

$$\|\nabla\mathcal{L}(\theta)\| = \mathcal{O}\left(\tilde{B} \cdot \sqrt{\mathscr{E}}\right),$$

*where $\mathscr{E} = \mathbb{E}_x[d_{\mathrm{TV}}(\hat{p}_\theta(\cdot|x), p(\cdot|x))]$ captures the ability of the transformer to solve the task and $\tilde{B}$ depends solely on the spectral norm of the model's parameters.*

We see that the term $\mathscr{E}$ captures the ability of the model to correctly solve the task. In the early iterations, the model behaves poorly, which imposes weak control on the gradient norm. This allows for large gradient norms and hence for bigger updates to the weights. The loss can vary rapidly, as can be seen by the steep descent of Fig. 5. During training, the model improves as can be seen by the increase in accuracy (both train and test), leading to a smaller $\mathscr{E}$ and strengthening the control on the norm. In parallel, the gradient

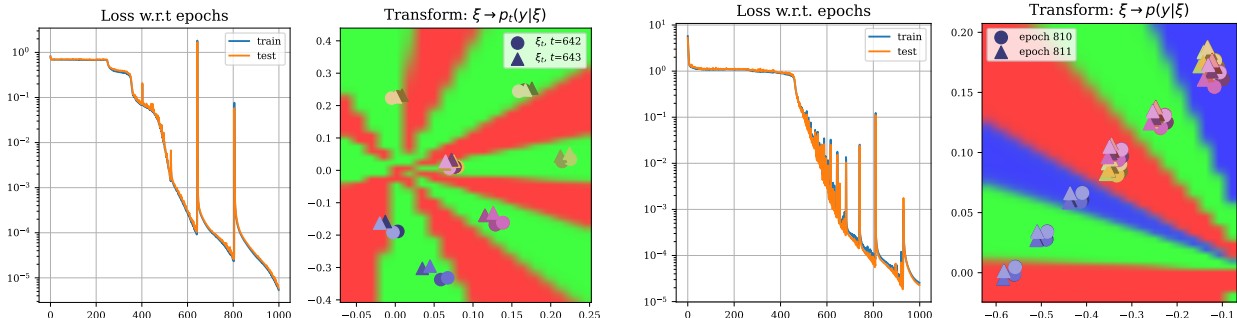

Figure 7: **High-curvature feed-forward**. Loss spikes (both left) resulting from a small change from one iteration to another in sequence embeddings that are close to the decision boundaries of the subsequent feedforward layer (both right).

updates are of smaller magnitude as well as the loss variations. This implies a less steep descent in the loss as observed in Fig. 5.

### 4.4 Occurrence of loss spikes

Better understanding why loss spikes occur is of great interest as they can be detrimental to the training stability, particularly with very large-scale transformers such as large language models (Chowdhery et al., 2023). While some know-how exists for mitigating the impact of peak losses, it is often experimentally motivated and requires a lot of computational resources to achieve operational success (Grattafiori et al., 2024). One interesting aspect of our sandbox is that it generates loss spikes that we can study quite precisely. Our visual inspection showcases two aspects leading to loss spikes: the high curvature of the RMS normalization layer near the origin, as well as the high curvature of multi-layer perceptrons with heavy weights, or with numerous small correlated weights.

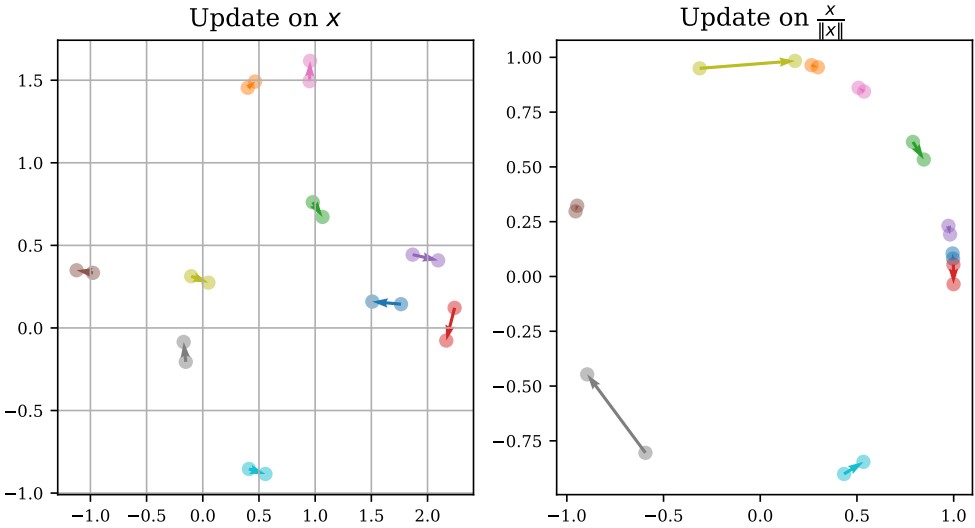

Figure 8: **High-curvature normalization**. Loss spikes are linked to the high curvature of internal network functions. A small update to an element $x$ can result in a substantial change to its normalized version $x / \|x\|$, significantly altering the network's subsequent behavior.

**High-curvature normalization.** Fig. 8 illustrates that a small modification to an element can result in a disproportionately large change in its normalized version. In theory, the gradient points towards directions that would reduce the training loss. However, considering a large step size in these directions could be counterproductive. This is especially true for functions with high curvature, such as the normalization layer $f(x) = x/\|x\|$ near the origin. At any point $x_t$, the gradient descent update rule suggests that one can update $x_{t+1}$ as $x_t - \eta_t u_t$ without changing $f(x_t)$, where $u_t = x_t/\|x_t\|$ and $\eta_t$ is the learning rate. This holds true only if the learning is small enough, $\eta_t < \|x_t\|$. When $x_t$ is close to zero, ensuring $\eta_t < \|x_t\|$ becomes challenging, particularly if the step size $\eta_t$ was predetermined by some scheduler. This behavior is related to the "edge-of-stability" phenomenon highlighted by Cohen et al. (2022), further theoretical insights being provided by Cabannes et al. (2024c). Interestingly, this analysis suggests removing some loss spikes by smoothing out the normalization layer. For example, consider using $f(x) = \sigma(\|x\|)x/\|x\|$ where $\sigma$ is a smooth function with $\sigma(0) = 0$ and $\sigma([1, \infty)) = \{1\}$. This demonstrates the usefulness of our visual sandbox in gaining insights and building intuition, which can then be validated on a larger scale in subsequent works.

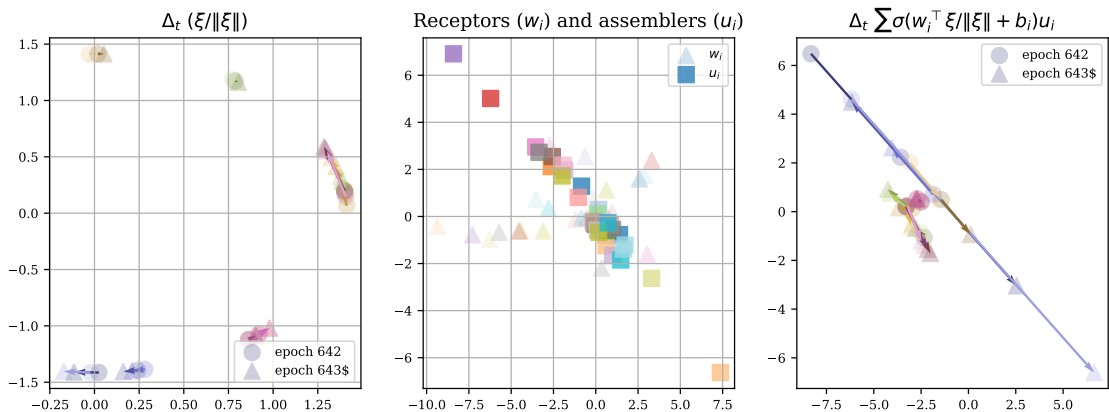

Figure 9: A small change in sequence embeddings (left) can lead to a big change in MLP response (right). This is due to heavy or small but heavily correlated assemblers (middle).

**High-curvature feed-forward.** Another source of loss spikes is illustrated in Fig. 7. They are due to the decision boundaries of the feedforward layer being quite close to the sequence embeddings, meaning that a small change in the sequence embedding can lead to a great change in their classification. This is again due to the high curvature of the MLP layer, as illustrated in Fig. 9. In particular, the heavy, or the small but heavily correlated, weights in the MLP cause the response $\zeta$ to vary highly as a function of $\zeta$. Once again, one can imagine different ways to regularize these types of loss spikes, with various regularization measures, or by ensuring that the capacity of the MLP is large enough for the MLP to avoid creating these heavy or highly correlated weights.

**Connection to gradient norms.** Loss spikes and high gradient norms appear in tandem as can be seen in Fig. 10, indicating that when a too-large step size deviates the model from its current small loss region, it is taken back to where it was with large updates. It motivates further study of the Hessian of the loss, which is promising in the literature (Foret et al., 2021; Gomes et al., 2025; Ilbert et al., 2024; Zhang et al., 2024c).

**Experiments in larger dimensions.** When the embedding dimension $d$ is bigger, we cannot represent the tokens in the plane. However, we can plot the evolution of other metrics, such as the loss, the gradient norms, or the activation sparsity. We replicate the experiments of Section 4.3 with $d > 2$ in Section A.5 and obtain similar findings, which are also coherent with the existing literature in more practical settings (Ilbert et al., 2024; Li et al., 2023b; Mirzadeh et al., 2024; Zhang et al., 2024c).

**Implication for large language models.** Ensuring the stability of training is key to reducing budget and computational costs. A common issue during pretraining is the occurrence of loss spikes (Chowdhery et al.,

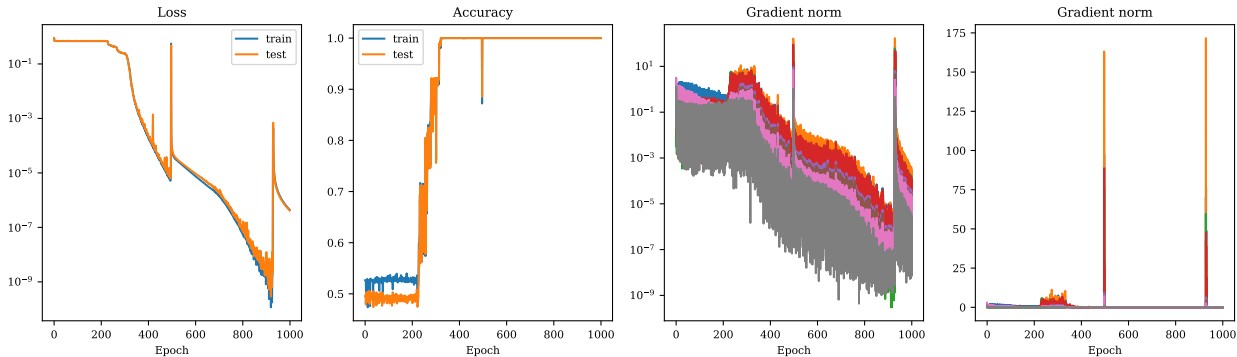

Figure 10: **Loss spikes and gradient norms.** Akin to Fig. 6 in mini-batch, this experiment makes the connection between gradient norm and loss spikes salient.

2024; Li et al., 2025; Marin, 2025, see), which can lead to loss divergence. Better understanding their root cause is crucial to reducing the cost of training large models, e.g., through the design of novel optimizers. Our findings suggest that reducing gradient norms and avoiding embeddings with small norms can help stabilize the training. This resonates with practical heuristics such as QK-Norm (Dehghani et al., 2023; Wortsman et al., 2024) or MuonClip (Kimi Team et al., 2025) that can be seen as a way to enforce small gradient norms by avoiding exploding attention logits, while imposing unit norm to embedding vectors, thus mitigating the high-curvature region of normalization layers, has been show to be beneficial for both the training stability and the sample efficiency (Loshchilov et al., 2025). Another promising avenue is the use of second-order algorithms such as Shampoo (Gupta et al., 2018), Muon (Jordan et al., 2024), and AdaFisher (Gomes et al., 2025), which allow for better control over the gradient variations.

## 5 Related work

**Mechanistic interpretability.** Neural networks are often seen as "black boxes" whose internal computations are hard to decipher. Researchers have developed various methods, such as extracting meaningful features from neural network activations (Fel et al., 2023), and assessing the impact of perturbations on model inputs (Fel et al., 2021; Koh & Liang, 2017), among others. Recently, the field of mechanistic interpretability has advocated for exposing the internal mechanisms of transformers to provide novel insights into their capabilities (Elhage et al., 2021; Olsson et al., 2022), by either relying on interventions on large models (Geva et al., 2023; Meng et al., 2022; Wang et al., 2023), or by conducting precise ablation studies in a controlled setup (Bietti et al., 2023; Cabannes et al., 2024a; Charton, 2022; Liu et al., 2022; Nanda et al., 2023). Our work aligns with the latter, aiming to make the internal behavior of transformers more explicit through carefully selected visualizations in controlled settings, with a focus on their training dynamics.

**Training dynamics in neural networks.** With non-convex losses and models of increasing sizes, understanding how models learning occurs is more and more difficult. One approach to address this issue is to employ mathematical abstractions, as seen with neural tangent kernels (NTK) Chizat et al. (2020); Jacot et al. (2018) and mean-field analysis (Chen et al., 2024; Mei et al., 2018), among others (Abbe et al., 2022; Ahn et al., 2024; Cabannes et al., 2024c). Obtaining formal results often requires deviating from the practical implementation (Ahn et al., 2023; Bietti et al., 2023; Boix-Adsera et al., 2023; Cabannes et al., 2024c; Geshkovski et al., 2023a; Jelassi et al., 2022; Mahankali et al., 2024; Tian et al., 2023). In our work, we keep the main transformer block components intact to ensure that our findings have practical implications.

### 5.1 Discussion

This paper aims to advance our understanding of the transformer's training dynamics through a detailed study in a controlled and interpretable setting. We show how transformers trained with gradient descent can solve the sparse modular addition by capturing invariants, as is often the case in NLP and reasoning.

Our visual sandbox allows us to monitor the evolution of tokens during training. We use it to uncover the stage-wise learning and the occurrence of loss spikes. By connecting our findings with more practical settings, we show the benefits of our sandbox to study complex behavior at a small scale. We hope it can be used to build intuition towards improving current training pipelines.

**Limitations.** Our work is a controlled study of a single transformer block, which allows both theory and experiments towards better understanding transformers training. In future work, we will aim to extend our visualization tools to larger models.

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

# Appendix

**Roadmap.**  We conduct additional experiments in Section A. In Section B, we discuss in detail the theoretical insights on the learning dynamics and provide the proofs in Section C.

## Table of Contents

# A   Experimental results

In this section, we present additional results related to the study of the circuits implemented by the transformer. Notably, we investigate the concrete realizations to solve the task, the sparsity of the model's activation along training, and the transferability of circuits.

**Implementation details.**   In our experiments, we trained our networks using $n = 2048 = 2^{11}$ data points, which were sampled uniformly with replacement from the $p^L$ possible sentences. We utilized the Adam optimizer with parameters $\beta_1 = 0.9$ and $\beta_2 = 0.999$ (Kingma & Ba, 2017), and we initialized the network weights using the default schemes provided by PyTorch (Paszke et al., 2019). Experiments were conducted on high-performance GPUs such as V100 and can be reproduced on a single device. The code will be open-sourced upon publication.

## A.1   Clustering heads implementations

We expect *clustering heads* to naturally appear during training, given that they are an efficient and simple way to solve the task. In practice, the sequence embeddings $\xi$ learned through gradient descent can be grouped in similar but different ways, which leads to several variations of the idealized head. These variations are categorized below.

**Faulty attention corrected by value.**   In many instances, we observe that the attention scores are not fully concentrated on the first five tokens but are compensated by the value matrix, which effectively collapses the embeddings of the spurious tokens that are attended to. This allows the sequence embeddings to remain invariant to the suffix of the sequence, as can be seen in Fig. 11.

**Partially learned invariants.**   Frequently, the sequence embeddings have not fully learned all the suffix and prefix invariants, resulting in more than six clusters of sequence embeddings[2]. Specifically, Fig. 12 illustrates a scenario where the sequence embeddings are not invariant to the value $x_6$, as evidenced by the positions of the blue squares $(0, 6)$ and $(1, 6)$ on the plot. They also lack invariance to permutations of the token in the first or fifth positions with another of the non-spurious tokens. This results in a sequence embedding that presents more clusters than the idealized model, leading to a greater number of connected decision regions in the feedforward layer.

**Fuzzy constructions.**   Occasionally, we encounter fuzzy constructions where the sequence embeddings are clustered according to unconventional patterns that nonetheless generalize to unseen data. Such a construction is presented in Fig. 13.

## A.2   Connectivity of neurons

Sparsity is a phenomenon of interest in many fields such as signal processing, neuroscience, and machine learning (Barth & Poulet, 2012; Chen et al., 1998; Mairal et al., 2009). Recent studies focused on the sparsity in deep neural network activations. In particular, Li et al. (2023a) showed that trained transformers have sparse activations and concluded that it was caused by the training dynamics rather than by a compact representation of the training data as commonly thought in computer vision and NLP. Mirzadeh et al. (2024) observed a similar phenomenon and showed how to leverage sparsity to reduce the inference cost of large language models. Inspired by this line of work, we study the activation sparsity of our model from a performance viewpoint. It should be noted that those works study deep transformers and identified that the sparsity increases with the depth while we only consider a one-layer transformer. In our setting, the activation sparsity can help us better understand how neurons connect to implement *clustering heads*.

**Successful models have dense activations.**   Following the framework from Li et al. (2023a), we recall that the activation sparsity corresponds to the percentage of non-zero entries of the feed-forward activation

---

[2]This corresponds to $\binom{k+p-1}{k}$ for $n = 2, k = 5$.

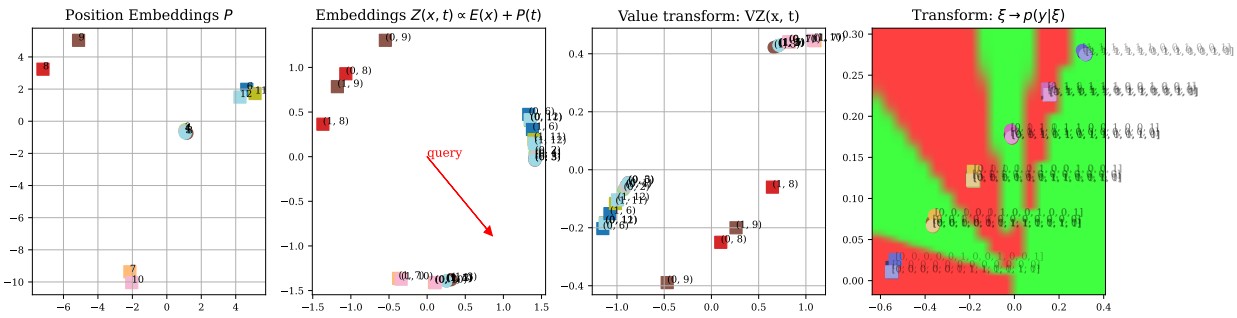

Figure 11: Faulty attention corrected by value collapse.

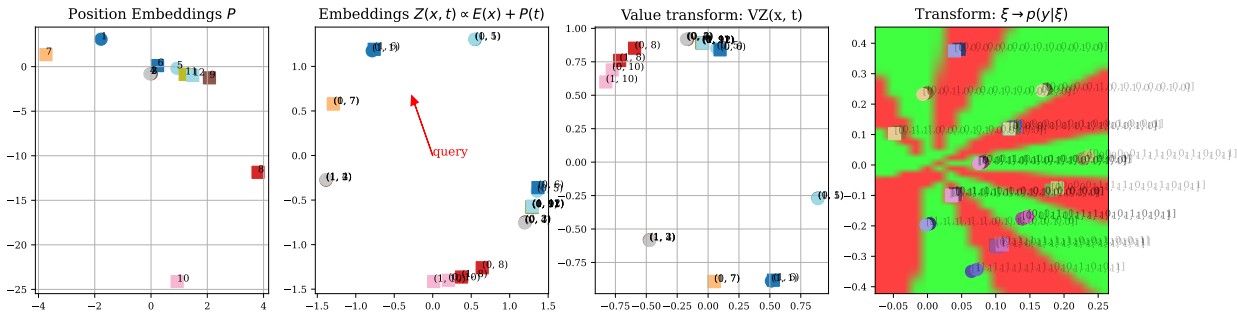

Figure 12: Embeddings having only learned some invariants.

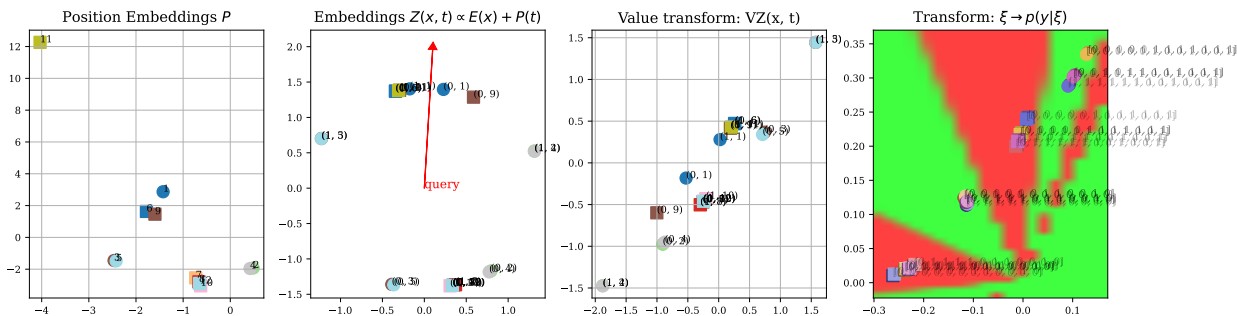

Figure 13: Fuzzy construction.

map. Without loss of generality, the feed-forward block is an MLP with weights $W_1, W_2$ and a non-linear activation $\sigma$ that outputs for any input $x$ a vector $z = W_1\sigma(W_2x)$. Formally, the activation sparsity is the percentage of non-zero entries in $\sigma(W_2x)$ and takes values in $[0, 1]$. In the classical setting with ReLU activation (Fukushima, 1969), this is equivalent to computing the percentage of non-negative neurons before the activation. However, some activations do not have non-negative outputs. This is the case of the SiLU (Elfwing et al., 2018), used in Llama models (Touvron et al., 2023a), and of the GeLU Hendrycks & Gimpel (2023) used in Falcon (Almazrouei et al., 2023), PaLM (Chowdhery et al., 2024), and in our transformer implementation. Instead of replacing such activations with a ReLU (Li et al., 2023a; Mirzadeh et al., 2024), we compute a smoothed sparsity. This is akin to using the $\ell_1$-norm, respectively the nuclear norm, instead of the $\ell_0$ quasi-norm, respectively the rank (Gribonval & Nielsen, 2003; Ilbert et al., 2024). as the percentage of entries with an absolute value lower than $\varepsilon > 0$. A sparsity of 1 means that all entries are $\varepsilon$-close to 0 (i.e., sparse activations), and a sparsity of 0 means that all entries are at least $\varepsilon$-away from 0 (i.e., dense activations).

**Evolution of the sparsity with performance.** To better understand the impact of sparsity, we train 20 independent models and display in Fig. 4 their sparsity after training for $\varepsilon \in [10^{-5}, 10^2]$ (the range is

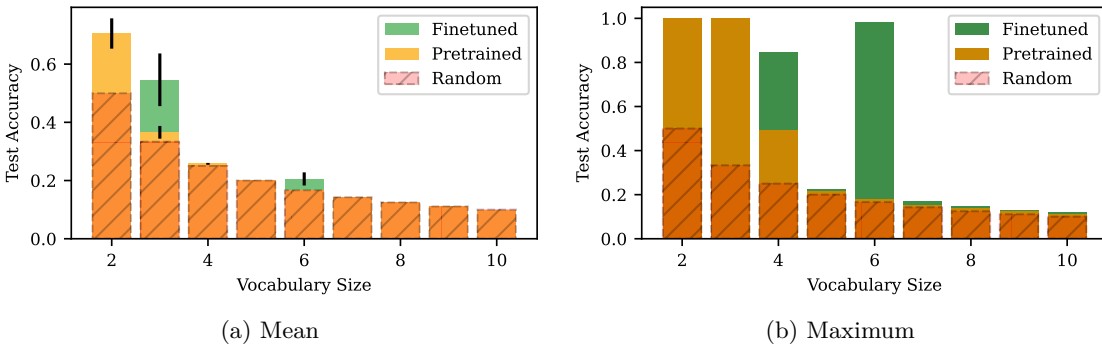

(a) Mean                                    (b) Maximum

Figure 14: Accuracies obtained from pretraining only with $p \in [2, 10]$ and finetuning with $p \in [3, 10]$ starting from $n = 2$ for various vocabulary sizes. **Left:** Averaged accuracy; **Right:** Maximum accuracy. Finetuned models display better performance than pretrained-only models.

chosen such that the sparsity reaches its extremal values 0 and 1). Given the task's difficulty, achieving an accuracy above 0.9 is a success; otherwise, it is a failure. On the left, we plot successful models in blue and failed ones in red. We observe a striking separation between successful and failed training. In the permissible range $[10^{-3}, 10]$, successful models tend to have less sparse activation than failed ones. To further study this phenomenon, we plot in the right subplot of Fig. 4 the evolution of the sparsity with $\varepsilon$, and here, the color indicates the models' test accuracy (the lighter the color, the better the model). We can see that the sparsity decreases as the performance increases. This explains the sharp transition between failure and success in the left subplot. These experiments seem to indicate that, contrary to images and textual data (Li et al., 2023a; Mirzadeh et al., 2024), the sparse modular addition problem needs the involvement of many neurons during inference, and hence requires non-sparse activations.

### A.3 Transferability of circuits

Among the lessons learned from training very large models is the importance of careful data engineering (Llama Team, 2024; Touvron et al., 2023b). Cabannes et al. (2024a) has highlighted that certain sources of data may facilitate the learning of invariances, while Abbe et al. (2023) discusses how data curation enables models to escape saddle points more quickly. These insights are consistent with the observations we made regarding our problem. In the sparse modular addition problem, the number of unique sequences is equal to $p^T$, which increases rapidly with both $p$ and $T$ and makes the problem quite hard to learn. In particular, when setting $T = 12$ and limiting the training set to $n = 2048$ data points, training for 1000 epochs does not result in any learning for $n > 4$.

**Impact on the performance.** As previously seen in Fig. 11, when training with $n = 2$, we often find circuits that capture both permutation and suffix invariants. These invariants generalize for any $n \in \mathbb{N}$ when $k$ and $T$ are fixed. Consequently, initializing models with these invariants makes learning the sparse modular addition problem much easier. This observation was made after conducting the following experiments: we first trained a model with sequences in $\mathbb{F}_n$ for $n = 2$ over 1000 epochs, before switching the dataset to sequences in $\mathbb{F}_n$ for $n = 3$ for another 1000 epochs. We found that this procedure significantly facilitates the learning of the sparse modular addition problem for $n = 3$, which we summarize in Fig. 14 where each bar plot is obtained by over 250 runs. Remarkably, we found that the only models that achieved 100% test accuracy were those that captured both the token and permutation invariances after the first 1000 epochs. Specifically, these were the models that created six sequence embedding clusters, as shown in Figures 1 and 11, rather than those depicted in Figures 12 or 13.

**Impact on the learned circuits.** Fig. 15b shows the final circuit found in one of our finetuning experiments. The training was initialized with the circuit in Fig. 11, after adding a token embedding to encode for $x = 2$, resulting in Fig. 15a. The final embeddings are not that far from the initial one, with the transformer having learned to mainly pay attention to the non-spurious tokens that are not equal to 2. It also pays some attention to 0 and 1 in positions $t = 7$ and $t = 10$. However, this faulty attention is corrected by the value matrix.



(a) Initialization with an additional token embedding for $n = 3$.

(b) Circuit learned after 1000 epochs of finetuning with $n = 3$.

Figure 15: *Transfer of clustering head.* The obtained circuit is akin to Fig. 11 where the value corrects the faulty attention.

Once again, the final configuration seems somewhat close to the initial one, as shown by the attention pattern reported in Fig. 2. This is consistent with the observations made in the previous subsections.

**Benefits of transfer.** Overall, our transfer experiment highlights the transferability of circuits and the usefulness of a curriculum in facilitating the learning of challenging tasks by inducing effective circuits through tasks that are easier to solve. Although our experiments are performed in two stages, we hypothesize that the same type of mechanism can explain the importance of data curation. We also note that curriculum learning has shown its benefits in other settings, such as semi-supervised learning and self-training (Cascante-Bonilla et al., 2021; Odonnat et al., 2024).

## A.4 Impact of the hyperparameters

In this section, we conduct ablation studies on the training hyperparameters. In Fig. 16a, we observe the high variability of the test accuracy for a fixed seed after 1000 epochs when varying the batch size, hidden dimension, learning rates, and MLP learning rates. Fig. 16b is similar with more regularity as it is averaged over 100 runs with different seeds. This study shows that the exit of saddle points is highly dependent on the hyperparameters.

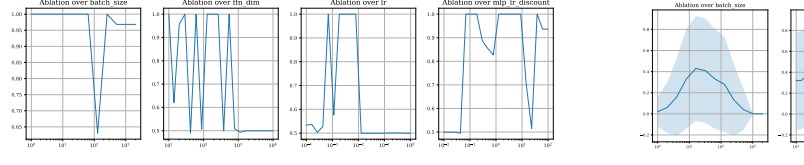

(a) Single run with 1000 epochs.

(b) Averaged over 100 runs with standard deviations.

Figure 16: *Ablation studies* regarding test accuracy as a function of batch size, hidden neurons, learning rates, and MLP learning rates discount factor for a single run with 1000 epochs. We ensured consistency in initial weights and batch designs when changing hyperparameters. We observe (mostly in Fig. 16b) some regularity in the effect of hyperparameters on the resulting test accuracy.

## A.5 Towards more practical setting

Relying on our visual sandbox, we studied several phenomena on the training dynamics of neural networks from the learning of the different parts of the network in Section 4.3 to the loss spikes occurring during the optimization in Section 4.4 through the efficiency of transfer learning in Section A.3. We also analyzed the connection between saddle points, loss spikes, and gradient norms in Sections 4.3 and 4.4 and the impact of the activation sparsity on the models' performance in Section A.2. The low-dimensional embeddings allowed us to pair loss profiles with the visualization of each layer to decipher the training behaviors. However, we note that the studied behaviors and the insights we obtain can be analyzed independently of the embedding dimension $d$.

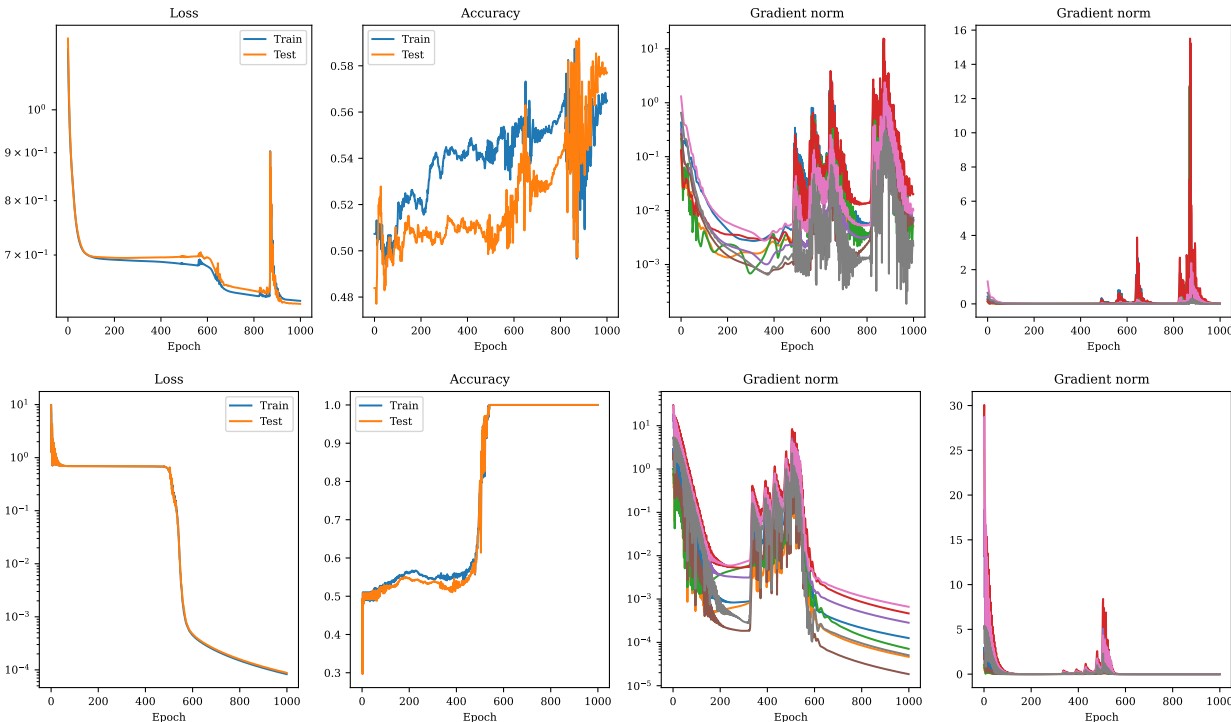

Figure 17: *Connection to saddle points.* This study is akin to Fig. 6 with $d = 3$ in the **top** plot and $d = 64$ in the **bottom** one. **From left to right:** Evolution of train and test losses, the corresponding accuracies, the evolution of gradient norms for each layer in log-scale, and the similar evolution in linear scale in full-batch. Akin to Fig. 6, we see that the learning phases studied in Section 4.3 appear in tandem with high gradient norms. This can be seen in the last subfigure, where the pics correspond to the loss drops and spikes.

**Motivation.** Taking an embedding size $d = 2$ was mainly motivated by visualization purposes, and we saw that such a setting was already challenging enough to observe training phenomena of interest in more practical settings. For the sake of self-consistency, we extend our experiments with higher-dimensional embeddings and show that our insights remain valid in more practical settings with $d > 2$.

**Two-phase learning and saddle points** In this section, we extend the experiments of Section 4.3 for $d > 2$. In Fig. 17, we consider $d = 3$ (top) and $d = 64$ (bottom). We obtain similar conclusions to those in the case $d = 2$ in Fig. 6 where the loss drops occur in tandem with high gradient norms for each layer, hinting at the exit of saddle points.

**Loss spikes and gradient norms** In this section, we extend the experiments of Section 4.4 for $d > 2$. In Fig. 18, we consider $d = 3$ (top) and $d = 64$ (bottom). We observe a similar behavior as in the case $d = 2$ in Fig. 10 where the loss spikes and gradient norm spikes occur at the same time. This again highlights the connection between gradient norms and loss spikes.

**Activation sparsity and performance** Finally, we extend the experiments of Appendix A.2 with an embedding size $d \in \{2, 3, 4, 8, 16, 32\}$ in Fig. 19. We first note that the higher the embedding size, the more the model succeeds at the task. Especially, as of $d = 8$, all the models are successful, i.e., they all achieve an accuracy higher than 0.9 (as defined in Section A.2). This is expected given that imposing low-dimensional embeddings limits the expressiveness and the generalization power of our model. It should be noted that this was one of the many challenges of our study: obtaining a generalizable neural network with embeddings in $R^2$ for a mathematical reasoning task such as the sparse modular addition problem. We obtain similar conclusions than in Section A.2 with successful models having denser activations.

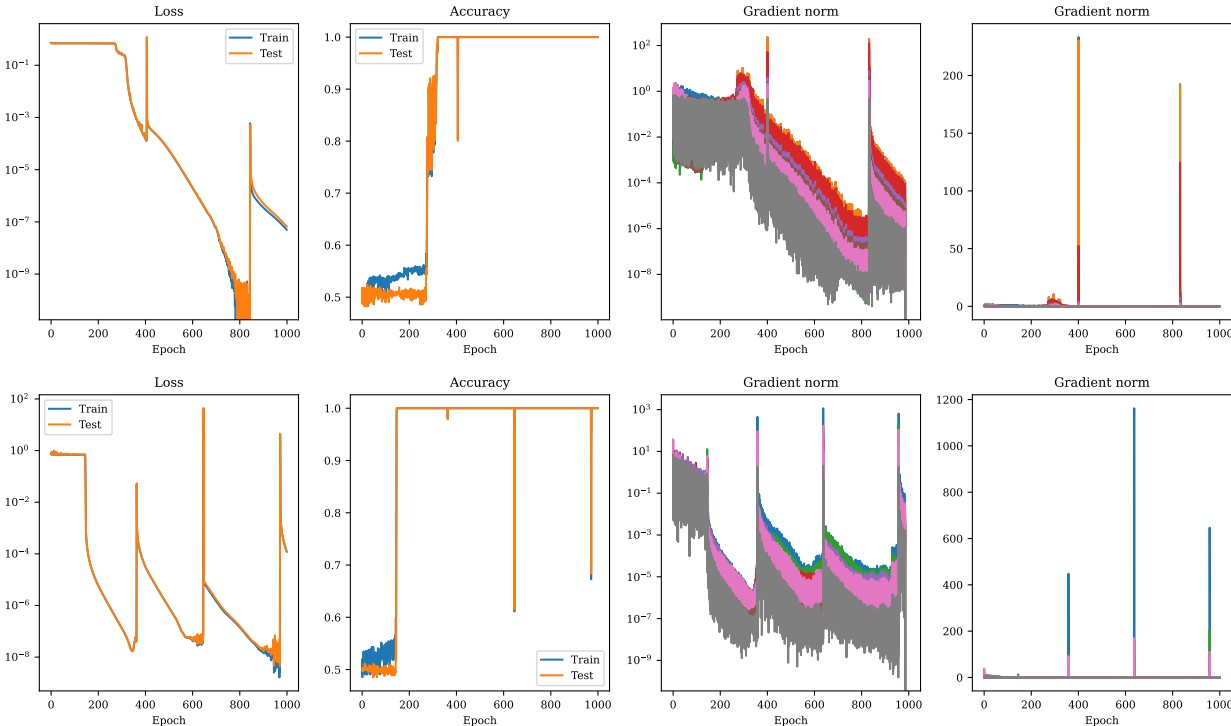

Figure 18: *Loss spikes and gradient norms.* This study is akin to Fig. 10 in mini-batch with $d = 3$ in the **top** plot and $d = 64$ in the **bottom** one. Again, we observe that the loss spikes and gradient norm spikes appear in tandem.

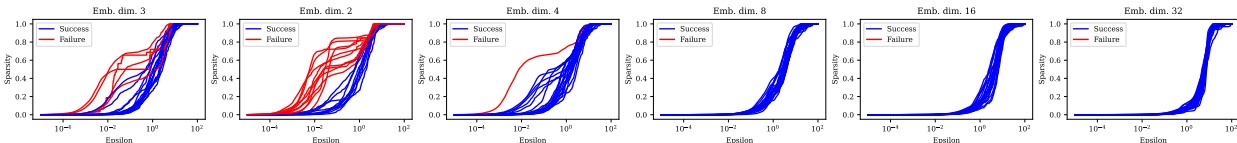

Figure 19: *Connection between performance and sparsity.* We display the evolution of the activation sparsity of 20 trained models with $\varepsilon \in [10^{-5}, 10^2]$ for $d \in \{2, 3, 4, 8, 16, 32\}$. Successful models (i.e., with test accuracy above 0.9) in blue have less sparse activation than failed models in red.

## B   Theoretical results

In this section, we detail our theoretical insights into the learning dynamics. It should be noted that all our derivations are conducted with an embedding dimension of $d$ and our transformer block contains the key components of its larger counterparts (see Section 2).

**Notations.**  In this section, we recall the notations used in our theoretical results and proofs. We denote $\{1, \cdots, N\}$ as $[N]$. The $i$-th row of a matrix $A \in \mathbb{R}^{n \times m}$ is denoted by $A_i$, its $j$-th column is denoted by $A_{.,j}$ and its transpose is denoted by by $A^\top$. By abuse of notation, we denote by $\|A\|$ the entry-wise $\ell_2$-norm of the matrix $A$, which amounts to computing the Euclidean norm of $A$ flattened. Note that we have $\|A\| = \|A\|_{\mathrm{F}} = \sqrt{\sum_{i=1}^{\min\{n,m\}} \sigma_i(A)^2}$, where $\|\cdot\|_{\mathrm{F}}$ is the Frobenius norm of $A$ and the $\sigma_i(A)$ are its singular values.The operator norm (also called spectral norm) of $A$ writes $\|A\|_{\mathrm{op}} = \sigma_{\max}(A)$, with $\sigma_{\max}(A)$ the largest singular value of $A$. The max norm of $A$ is denoted by $\|A\|_\infty = \max_{ij} |A_{ij}|$. The identity matrix of size $n$ is denoted by $I_n \in \mathbb{R}^{n \times n}$. The indicator function is denoted by $\mathbb{1}$ and verifies $\mathbb{1}\{a = b\}$ with value 1 if $a = b$ and 0 otherwise. The transpose of a vector $x$ is denoted by $x^\top$, and its Euclidean norm is denoted by

$\|x\| = \sqrt{x^\top x}$. The total variation distance between two probability distributions $p, q$ is denoted by $d_{\mathrm{TV}}(p, q)$. For ease of notation, we define the following quantities for $\xi \in \mathbb{R}^d, W \in \mathbb{R}^{h \times d}, z \in \mathbb{R}^{d \times N}, q \in \mathbb{R}^d$ and the GeLU function $\sigma \colon \mathbb{R} \to R$:

$$\bar{\xi} = \frac{\xi}{\|\xi\|} \in \mathbb{R}^d, \tag{1}$$

$$A_z = \mathrm{diag}\left(\mathrm{softmax}(\frac{z^\top q}{\sqrt{d}})\right) \in \mathbb{R}^{N \times N}, \tag{2}$$

$$\Sigma_{\bar{\xi}} = \mathrm{diag}(\sigma'(W\bar{\xi})) \in \mathbb{R}^{h \times h}. \tag{3}$$

$$\Delta_z = \frac{z}{\sqrt{d}}\left[I_N - \mathrm{softmax}(\frac{z^\top q}{\sqrt{d}})\mathbb{1}_N^\top\right]. \tag{4}$$

## B.1 Gradients derivations

In this section, we derive the gradient of the loss with respect to the neural network's weights. To avoid cumbersome derivations that do not fundamentally alter the analysis, and since we are primarily interested in the dynamics at work in the attention and feed-forward layers, we assume that the embedding matrices $E$ and $P$ are randomly initialized and do not participate in the optimization. It also concerns the unembedding matrix $W_U$ thanks to weight tying. This is a common assumption in prior works studying the training dynamics of transformers (Bietti et al., 2023; Cabannes et al., 2024b,c; Nichani et al., 2024). To derive the loss gradients, we first recall the quantities involved and reformulate the loss.

1. **Input**: $x = [x_1, \ldots, x_L] \in (\mathbb{F}_p)^L$,
2. **Embedding**: $z_t := Z(x_t, t) := \frac{E(x_t) + P(t)}{\|E(x_t) + P(t)\|} \in \mathbb{R}^d$ with $P, E$ learnable,
3. **Attention**: $\xi := (Vz)\,\mathrm{softmax}(\frac{z^\top q}{\sqrt{d}}) \in \mathbb{R}^d$ with $V \in \mathbb{R}^{d \times d}, q \in \mathbb{R}^d$ randomly fixed following (Bietti et al., 2023),
4. **Feed-forward**: $\psi := \xi + \sum_{i \in [h]} u_i \cdot \sigma\left(\frac{w_i^\top \xi}{\|\xi\|}\right) = \xi + U\sigma\left(\frac{W\xi}{\|\xi\|}\right) \in \mathbb{R}^d$ where $U \in \mathbb{R}^{d \times h}$ has columns $(u_i)_{i=1}^h$, $W \in \mathbb{R}^{h \times d}$ has rows $(w_i^\top)_{i=1}^h$ and $U, W$ are learnable,
5. **Unembedding**: $\zeta = (E(v)^\top \psi)_{v \in \mathbb{F}_p} = W_U \psi \in \mathbb{R}^n$ where $W_U \in \mathbb{R}^{p \times d}$ has rows $E(v)^T$.

**Gradient of the training loss.** The vector $\theta$ encompasses all the model's parameters. More specifically, $\theta$ can be seen as the (flattened) concatenation of $q \in \mathbb{R}^d, V \in \mathbb{R}^{d \times d}, W \in \mathbb{R}^{h \times d}, U \in \mathbb{R}^{d \times h}$, i.e., as a vector of $R^m$ with $m = d(1 + d + 2h)$. We recall that the cross-entropy loss can be reformulated as

$$\mathcal{L}(\theta) := \mathbb{E}_{(x,y) \sim q}[\ell(\theta; x, y],$$

where we have

$$\ell(\theta; x, y) = -\sum_{i=1}^p \mathbb{1}\{y = i\} \log(\frac{\exp(\zeta_i)}{\sum_{n=1}^p \exp(\zeta_n)}) = \log(\sum_{n=1}^p \exp(\zeta_n)) - \sum_{i=1}^p \mathbb{1}\{y = i\} \cdot \zeta_i. \tag{5}$$

We can now proceed with computing the gradient of the loss function. The following lemma, whose proof is deferred to Section C.1, gives the closed-form expression of the gradient.

**Lemma 2** (Full statement of Lemma 1). *Let* $\Delta_z = \frac{z}{\sqrt{d}}\left[I_N - \mathrm{softmax}(\frac{z^\top q}{\sqrt{d}})\mathbb{1}_N^\top\right] \in \mathbb{R}^{d \times N}$, $A_z = \mathrm{diag}\left(\mathrm{softmax}(\frac{z^\top q}{\sqrt{d}})\right) \in \mathbb{R}^{N \times N}$, $\Sigma_{\bar{\xi}} = \mathrm{diag}(\sigma'(W\bar{\xi})) \in \mathbb{R}^{h \times h}$, $M = \frac{1}{\|\xi\|}\left(I_d - \bar{\xi}\bar{\xi}^\top\right) \in \mathbb{R}^{d \times d}$ *and* $Q = U\Sigma_{\bar{\xi}}WM \in \mathbb{R}^{d \times d}$. *Assume that the embedding weights* $E, P$ *are not trainable and let* $[\cdot]$ *indicate vector concatenation where matrices are flattened into vectors whenever needed. The gradient of the*

*loss function is a vector $\nabla_\theta \mathcal{L}(\theta) \in \mathbb{R}^m$ with $m = d(1 + d + 2h)$ that writes*

$$\nabla \mathcal{L}(\theta) = \sum_{j=1}^{p} \mathbb{E}_x[(\hat{p}_\theta(y = j|x) - p(y = j|x)) \cdot \nabla_\theta \zeta_j],$$

*where*

$$\begin{cases} \nabla_\theta \zeta_j &= [\nabla_q \zeta_j, \nabla_V \zeta_j, \nabla_W \zeta_j, \nabla_U \zeta_j] \in \mathbb{R}^m, \\[2mm] \nabla_q \zeta_j &= \Delta_z A_z (Vz)^\top (I_d + Q^\top) E(j) \in \mathbb{R}^d, \\[2mm] \nabla_V \zeta_j &= (I_d + Q^\top) E(j) \left[ z \operatorname{softmax}(\frac{z^\top q}{\sqrt{d}}) \right]^\top \in \mathbb{R}^{d \times d}, \\[2mm] \nabla_W \zeta_j &= \Sigma_{\bar{\xi}} U^\top E(j) \bar{\xi}^\top \in \mathbb{R}^{h \times d}, \\[2mm] \nabla_U \zeta_j &= E(j) \sigma(W\bar{\xi})^\top \in \mathbb{R}^{d \times h}. \end{cases}$$

We first notice the importance of the token embeddings since each gradient depends (linearly) on the $E(j)$. We can then see that the attention gradients are both controlled by $(I_d + Q^\top) E(j)$. We note that $M = \frac{1}{\|\xi\|}(I_d - \bar{\xi}\bar{\xi}^\top)$ projects any vector into $\bar{\xi}$'s orthogonal complementary space, which is included in the unit-sphere of $R^d$ denoted by $\mathbb{S}^d = \{x \in \mathbb{R}^d \text{ s.t. } \|x\| = 1\}$. Similar orthogonal projections appear in the derivations of Tian et al. (2023) and are used to mimic normalization layers in Geshkovski et al. (2023a). With $M$ symmetric, $Q^\top = M(U\Sigma_{\bar{\xi}}W)^\top$ simply perturbs the vector $E(j)$ before projecting it in the sphere and $(I_d + Q^\top) E(j)$ can be seen as a perturbed residual connection. Finally, we observe that the gradients of the feed-forward weights have a linear dependency on $U, W$ which seems to allow implementing the decision boundaries once the sequence embeddings are clustered (see Section 4.3). Given that the gradient formulae in Lemma 2 are not straightforward to analyze due to the high dimension and the cross-dependency of the terms, we do not attempt to study the gradient updates, noting that a precise analysis is an important theoretical question for future work.

**Remark B.1** (Faithful to practice). *The derivations to obtain the gradients in closed form are somewhat involved due to the complexity of the network since we do not theoretically simplify the transformer architecture or the data contrary to most prior works that either model associative memories (Bietti et al., 2023; Cabannes et al., 2024b,c; Nichani et al., 2024), consider orthogonal token embeddings (Bietti et al., 2023; Tian et al., 2023), linearize the attention (Ahn et al., 2023; Mahankali et al., 2024; Nichani et al., 2024; Zhang et al., 2024b), discard residual connections (Geshkovski et al., 2023a; Jelassi et al., 2022; Tian et al., 2023) or the feed-forward block (Ahn et al., 2023; Edelman et al., 2024; Geshkovski et al., 2023a; Jelassi et al., 2022; Mahankali et al., 2024; Tian et al., 2023). Moreover, we preserve the classification setting with the cross-entropy loss used in practice instead of considering the regression case with a square loss (Chistikov et al., 2023; Pesme & Flammarion, 2023). While this ensures our analysis stays close to the practice, it also comes with technical challenges.*

## B.2 Stationary points characterization

The neural networks considered in this work are learned via gradient descent, which converges to one of its fixed points (Nesterov, 2014). Such points are *stationary points* (or *critical points*) of the loss function (Chi et al., 2019b), formally defined below.

**Definition 1** (Stationary points). *The stationary points of $\mathcal{L}$ are the solutions of the Euler equation*

$$\nabla \mathcal{L}(\theta) = 0. \tag{6}$$

> *A stationary point $\bar{\theta}$ can be either a local minimum, a local maximum, or a saddle point of $\mathcal{L}$.*

In deep learning, since most objectives are non-convex and in high dimensions, minima[3] are rarely global, and the gradient descent likely converges to a local minimum or a saddle point. In particular, saddle points are prevalent in deep neural networks' loss landscape (Dauphin et al., 2014), which can hinder the optimization.

**Remark B.2** (Beyond stationary points)**.** *We note that distinguishing the critical points between local minimum, local maximum, or saddle points involves computing the Hessian of the training loss (Chi et al., 2019b), which is beyond the scope of this work.*

As a first step towards better understanding the training dynamics, we show in the following theorem that *clustering heads* can be learned by gradient descent. The proof is deferred to Section C.2.

---

**Proposition 3** (Full statement of Proposition 1)**.** *Let $\Delta_z = \frac{z}{\sqrt{d}}\left[I_N - \text{softmax}(\frac{z^\top q}{\sqrt{d}})\mathbb{1}_N^\top\right] \in \mathbb{R}^{d \times N}$, $A_z = \text{diag}\left(\text{softmax}(\frac{z^\top q}{\sqrt{d}})\right) \in \mathbb{R}^{N \times N}$, $\Sigma_{\bar{\xi}} = \text{diag}(\sigma'(W\bar{\xi})) \in \mathbb{R}^{h \times h}$, $M = \frac{1}{\|\xi\|}\left(I_d - \bar{\xi}\bar{\xi}^\top\right) \in \mathbb{R}^{d \times d}$, $Q = U\Sigma_{\bar{\xi}}WM \in \mathbb{R}^{d \times d}$, $\bar{Q} = \bar{U}\Sigma_{\bar{\xi}}\bar{W}M \in \mathbb{R}^{d \times d}$ and $\mathcal{C}_x = \sum_{j=1}^{p}(\hat{p}_\theta(y = j|x) - p(y = j|x))E(j)$. The stationary points $\theta$ of $\mathcal{L}$ verify*

$$\mathbb{E}_x\left[\Delta_z A_z (Vz)^\top \left(I_d + Q^\top\right)\mathcal{C}_x\right] = 0,$$

$$\mathbb{E}_x\left[\left(I_d + Q^\top\right)\mathcal{C}_x\left[z\,\text{softmax}(\frac{z^\top q}{\sqrt{d}})\right]^\top\right] = 0,$$

$$\mathbb{E}_x\left[\Sigma_{\bar{\xi}}U^\top \mathcal{C}_x\bar{\xi}^\top\right] = 0,$$

$$\mathbb{E}_x\left[\mathcal{C}_x\sigma(W\bar{\xi})^\top\right] = 0.$$

(7)

*Moreover, Eq. (7) holds for transformers that implement clustering heads. This implies that such circuits can be learned by gradient descent.*

---

Proposition 3 allows us to access the final configuration of the model in the form of a system of equations. It involves the main quantities of the problem: the trainable weights $q, V, W, U$, the embedding tokens $E(j)$, and the training data $(x, y)$. A particularly interesting observation is the linear dependency of each equation on the term $\mathcal{C}_x$. This term captures the model's prediction discrepancy with the ground truth through the $\hat{p}_\theta(y = j|x) - p(y = j|x)$ and plays an important role in the proof. In particular, once the stationary points are characterized, showing that clustering heads can be learned by gradient descent is rather straightforward using the $\mathcal{C}_x$. In addition, it shows that the natural way we proposed observed to solve the sparse modular addition problem also qualifies as a fixed-point of gradient descent.

**Remark B.3** (Other pathways)**.** *While Proposition 3 shows that clustering heads are viable fixed points of the gradient descent, it does not provide conditions to ensure it is reached, nor qualify them as local minima, global minima, or saddle points. This interesting research direction is left for future work. We note that our experimental investigation suggests that clustering heads solve the sparse modular addition problem, but there may be other pathways to do so, too. In addition, as discussed in Section 4.2, the pathways learned in practice can differ from the idealized circuit, although the variations still capture most of the invariants of the problem.*

### B.3  Bounding the gradients

In Section 4.3, we connect some of our findings with the evolution of gradient norms along the training. Leveraging the closed-form expressions given in Lemma 2, we can study upper-bound gradient norms to elucidate their dependency on the parameters and data. The following proposition, whose proof is deferred to Section C.3, makes those upper bounds explicit.

---

[3]We omit the mention of maxima since we are mostly interested in minimization problems.

**Proposition 4** (Full statement of Proposition 2). *Assume the token embedding is bounded, i.e., there exists $B > 0$ such that $\|E\|_\infty \leq B$. We introduce $\mathscr{E} = \mathbb{E}_x[d_{\mathrm{TV}}(\hat{p}_\theta(\cdot|x), p(\cdot|x))]$ that captures the ability of the model to solve the sparse modular addition problem task. We have*

$$\|\nabla\mathcal{L}(\theta)\| = \mathcal{O}\left(\tilde{B} \cdot \sqrt{\mathscr{E}}\right),$$

*where*

$$\tilde{B} = B\sqrt{\|V\|_{\mathrm{op}}\left[1 + \|U\|_{\mathrm{op}}\|W\|_{\mathrm{op}}^2\right]^2 + \left[1 + \|U\|_{\mathrm{op}}\|W\|_{\mathrm{op}}\right]^2 + \|U\|_{\mathrm{op}}^2 + \|W\|_{\mathrm{op}}^2}.$$

This result is of interest for two main reasons. The first one is that it can be used to understand the loss profiles identified in Section 4.3. Notably, we see that the term $\mathscr{E}$ captures the ability of the model to correctly solve the task. In the early iterations, the model behaves poorly, which imposes weak control on the gradient norm. This allows for large gradient norms and hence for bigger updates to the weights. The loss can vary rapidly, as can be seen by the steep descent of Fig. 5. Along with the training, the model improves, as can be seen by the increase in accuracy (both train and test). This leads to a smaller $\mathscr{E}$ and strengthens the control on the norm. In parallel, the gradient updates are of smaller magnitude, as well as the loss variations. This goes along a less steep descent in the loss as observed in Fig. 5. The second benefit of Proposition 4 is that one could derive sufficient conditions on the term related to the weight matrices and on the term related to classification to achieve an approximate critical point (Chi et al., 2019b), that is parameters $\theta$ such that the gradient verifies $\|\nabla\mathcal{L}(\theta)\| \leq \varepsilon$ for some $\varepsilon > 0$. Since finding local approximate minima can be faster (and more beneficial) than finding critical points (Agarwal et al., 2017), practitioners could derive optimization algorithms that impose an appropriate bound on the two terms of Proposition 4 instead of using gradient descent. These interesting considerations are left for future work.

## C   Proofs

In this section, we provide detailed proof of our theoretical results.

### C.1   Proof of Lemma 1

We detail below the proof of Lemma 2.

*Proof.* The proof consists of three main steps:

1. Expressing the gradient of the loss with respect to the gradients of the attention embeddings,
2. Deriving the gradients for each trainable weight,
3. Putting everything together.

We first formulate the gradient of the loss with respect to the gradient of the attention embeddings. The next lemma provides the formula.

**Lemma 3** (Gradient compact formulation). *Denoting the model's trainable parameters by $\theta \in \mathbb{R}^m$ and recalling that the attention embedding $\xi \in \mathbb{R}^d$ can be seen as a function of $\theta$, we have*

$$\nabla_\theta\mathcal{L}(\theta) = \sum_{j=1}^{p}\mathbb{E}_x[(\hat{p}_\theta(y = j|x) - p(y = j|x)) \cdot \nabla_\theta\zeta_j] \tag{8}$$

*Proof.* Noting that the logits can be written as $\zeta = g(\theta; x, y)$ for some $g$, the loss[4] can then be written as $\ell(\theta; x, y) = \ell(\zeta) = \ell(g(\theta; x, y))$. We drop the $(x, y)$ letter when there is no possible ambiguity to avoid

---

[4]We keep denoting the loss by $\ell$ by abuse of notation.

cumbersome notations. Using the chain rule, with $D$ denoting the Jacobian operator, we have

$$
\begin{aligned}
\nabla_\theta \mathcal{L}(\theta) &= \mathbb{E}_{(x,y)\sim q}[\nabla_\theta \ell(\theta)] \\
&= \mathbb{E}_{(x,y)\sim q}[\nabla_\theta \ell(g(\theta))] \\
&= \mathbb{E}_{(x,y)\sim q}\left[(D_\theta g(\theta))^\top (D_{g(\theta)}\ell(g(\theta)))^\top\right] \\
&= \mathbb{E}_{(x,y)\sim q}[\nabla_\theta g(\theta)\nabla_\zeta \ell(\zeta)].
\end{aligned}
$$

By definition, we have $\nabla_\theta g(\theta) = (\nabla_\theta \zeta_1,\ldots,\nabla_\theta \zeta_p)$ and $\nabla_\zeta \ell(\zeta) = (\frac{\partial \ell}{\partial \zeta_j}(\zeta)) \in \mathbb{R}^p$. Hence, the matrix-vector product writes

$$
\begin{aligned}
(\nabla_\theta g(\theta))\nabla_\zeta \ell(\zeta) &= \sum_{j=1}^p \frac{\partial \ell}{\partial \zeta_j}(\zeta) \cdot [\nabla_\theta g(\theta)]_{\cdot,j} \\
&= \sum_{j=1}^p \frac{\partial \ell}{\partial \zeta_j}(\zeta) \cdot \nabla_\theta \zeta_j.
\end{aligned}
$$

Using Eq. (5), we have

$$
\begin{aligned}
\frac{\partial \ell}{\partial \zeta_j}(\zeta) &= \frac{\exp(\zeta_j)}{\sum_{n=1}^p \exp(\zeta_n)} - \mathbb{1}\{y = j\} \\
&= \mathrm{softmax}(\zeta)_j - \mathbb{1}\{y = j\} \\
&= \hat{p}_\theta(y = j|x) - \mathbb{1}\{y = j\},
\end{aligned}
$$

Putting everything together, we obtain

$$
\begin{aligned}
\nabla_\theta \mathcal{L}(\theta) &= \mathbb{E}_{(x,y)\sim p}\left[\sum_{j=1}^p (\hat{p}_\theta(y = j|x) - \mathbb{1}\{y = j\}) \cdot \nabla_\theta \zeta_j\right] \\
&= \sum_{j=1}^p \mathbb{E}_{(x,y)\sim p}[(\hat{p}_\theta(y = j|x) - \mathbb{1}\{y = j\}) \cdot \nabla_\theta \zeta_j] \\
&= \sum_{j=1}^p \mathbb{E}_x[\mathbb{E}_y[(\hat{p}_\theta(y = j|x) - \mathbb{1}\{y = j\}) \cdot \nabla_\theta \zeta_j|x]],
\end{aligned}
$$

where the first equality comes from the linearity of the expectation and the second stems from the definition of the conditional expectation. As $\hat{p}_\theta(y = j|x)$ and $\nabla_\theta \zeta_j$ do not depend on the random variable $y$, we have

$$
\begin{aligned}
&\mathbb{E}_y[(\hat{p}_\theta(y = j|x) - \mathbb{1}\{y = j\}) \cdot \nabla_\theta \zeta_j|x] \\
&= (\hat{p}_\theta(y = j|x) - \mathbb{E}_y[\mathbb{1}\{y = j\}|x]) \cdot \nabla_\theta \zeta_j \\
&= (\hat{p}_\theta(y = j|x) - p(y = j|x)) \cdot \nabla_\theta \zeta_j,
\end{aligned}
$$

since we have $x, y \sim p$. In summary, the gradient of the loss writes

$$
\nabla_\theta \mathcal{L}(\theta) = \sum_{j=1}^p \mathbb{E}_x[(\hat{p}_\theta(y = j|x) - p(y = j|x)) \cdot \nabla_\theta \zeta_j].
$$

$\square$

Lemma 3 implies that deriving the $\nabla_\theta \zeta_j = \left(\frac{\partial \zeta_j}{\partial \theta_n}\right)_{n=1}^m$ allows us to obtain the final gradient formula. The following lemmas provide for a fixed $j$ the gradients of $\xi_j$ with respect to the different parts of the neural networks, namely the feed-forward weights $U$ and $W$ and the attention weights $q$ and $V$.

### C.1.1 Attention derivations

In this section, we derive the gradients with respect to the attention weights.

**Attention query.** The following lemma provides the formula of $\nabla_q \xi_j$.

> **Lemma 4** (Query). *Let* $\Delta_z = \frac{z}{\sqrt{d}}\left[I_N - \text{softmax}(\frac{z^\top q}{\sqrt{d}})\mathbb{1}_N^\top\right] \in \mathbb{R}^{d \times N}$ *following Eq.* (4), $A_z = \text{diag}\left(\text{softmax}(\frac{z^\top q}{\sqrt{d}})\right) \in \mathbb{R}^{N \times N}$ *following Eq.* (2), $\Sigma_{\bar\xi} = \text{diag}(\sigma'(W\bar\xi)) \in \mathbb{R}^{h \times h}$ *following Eq.* (3), $M = \frac{1}{\|\xi\|}\left(I_d - \bar\xi\bar\xi^\top\right) \in \mathbb{R}^{d \times d}$ *following Eq.* (15) *and* $Q = U\Sigma_{\bar\xi}WM \in \mathbb{R}^{d \times d}$. *The entries of* $\nabla_\theta \zeta_j$ *associated to the value* $V$ *are the entries of* $\nabla_q \zeta_j \in \mathbb{R}^d$ *where we have*
>
> $$\nabla_q \zeta_j = \Delta_z A_z [(I_d + Q)Vz]^\top E(j).$$

*Proof.* We want to compute the $\frac{\partial \zeta_j}{\partial \theta_n}$ for the $\theta_n$ that are entries of $q$. We first note that

$$\frac{\partial \zeta_j}{\partial \theta_n} = \sum_{l=1}^{p} (W_U)_{jl} \frac{\partial \psi_l}{\partial \theta_n}, \tag{9}$$

where we have thanks to weight tying, $(W_U)_{jl} = E(j)_l$ is fixed. Moreover, we have

$$\psi_l = \xi_l + \sum_{i=1}^{h} (u_i)_l \sigma(w_i^\top \bar\xi), \tag{10}$$

where we recall that $\bar\xi = \frac{\xi}{\|\xi\|}$ following Eq. (1). Considering $n$ such that $\theta_n = q_a$ for some $a \in [d]$, we have

$$\frac{\partial \psi_l}{\partial \theta_n} = \frac{\partial \psi_l}{\partial q_a} = \frac{\partial \xi_l}{\partial q_a} + \frac{\partial}{\partial q_a}\left(\sum_{i=1}^{h}(u_i)_l \sigma(w_i^\top \bar\xi)\right) = \underbrace{\frac{\partial f_1(q_a)}{\partial q_a}}_{\text{LHS}} + \underbrace{\sum_{i=1}^{h}(u_i)_l \frac{\partial f_2^i(q_a)}{\partial q_a}}_{\text{RHS}}, \tag{11}$$

with

$$\begin{cases} f_1: & \mathbb{R} \to \mathbb{R}, \quad q_a \mapsto \xi_l \\ f_2^i: & \mathbb{R} \to \mathbb{R}, \quad q_a \mapsto (\sigma \circ f_3^i \circ f_4^i \circ f_5^i)(q_a) = \sigma(w_i^\top \bar\xi) \\ f_3^i: & \mathbb{R}^d \to \mathbb{R}, \quad \bar\xi \mapsto w_i^\top \bar\xi \\ f_4^i: & \mathbb{R}^d \to \mathbb{R}^d, \quad \xi \mapsto \bar\xi = \frac{\xi}{\|\xi\|} \\ f_5^i: & \mathbb{R} \to \mathbb{R}^d, \quad q_a \mapsto \xi. \end{cases}$$

We proceed in two steps and first compute the LHS of Eq. (11).

**LHS of Eq.** (11)**:** we first note that

$$\xi_l = \sum_{t=1}^{N} (Vz)_{lt} \, \text{softmax}\left(\frac{z^\top q}{\sqrt{d}}\right)_t.$$

This leads to

$$\frac{\partial f_1(q_a)}{\partial q_a} = \frac{\partial \xi_l}{\partial q_a} = \sum_{t=1}^{N} (Vz)_{lt} \frac{\partial s_t}{\partial q_a}, \tag{12}$$

where

$$s_t = \text{softmax}\left(\frac{z^\top q}{\sqrt{d}}\right)_t = \frac{\exp\left(\frac{1}{\sqrt{d}}\sum_{o=1}^{d} z_{ot}q_o\right)}{\sum_{t'=1}^{N}\exp\left(\frac{1}{\sqrt{d}}\sum_{o=1}^{d} z_{ot'}q_o\right)}. \tag{13}$$

This can be rewritten as

$$s_t = \frac{\exp\left(\frac{1}{\sqrt{d}}z_{at}q_a + \frac{1}{\sqrt{d}}\sum_{o\neq a}z_{ot}q_o\right)}{\sum_{t'=1}^N \exp\left(\frac{1}{\sqrt{d}}z_{at'}q_a + \frac{1}{\sqrt{d}}\sum_{o\neq a}z_{ot'}q_o\right)} = \frac{g(q_a)}{\tilde{g}(q_a)},$$

where

$$\begin{cases} g\colon x \to \exp\left(\frac{1}{\sqrt{d}}z_{at}x + \frac{1}{\sqrt{d}}\sum_{o\neq a}z_{ot}q_o\right) \\ \tilde{g}\colon x \to \sum_{t'=1}^N \exp\left(\frac{1}{\sqrt{d}}z_{at'}x + \frac{1}{\sqrt{d}}\sum_{o\neq a}z_{ot'}q_o\right) \end{cases}.$$

This leads to

$$\frac{\partial s_t}{\partial q_a} = \frac{g'(q_a)\tilde{g}(q_a) - \tilde{g}'(q_a)g(q_a)}{[\tilde{g}(q_a)]^2}.$$

By simple application of the chain rule, we have

$$g'(q_a) = \frac{1}{\sqrt{d}}z_{at}\exp\left(\frac{1}{\sqrt{d}}\sum_{o=1}^d z_{ot}q_o\right)$$
$$= \frac{1}{\sqrt{d}}z_{at}\exp\left(\frac{1}{\sqrt{d}}z_{.,t}^\top q\right).$$

Similarly, we have

$$\tilde{g}'(q_a) = \sum_{t'=1}^N \frac{1}{\sqrt{d}}z_{at'}\exp\left(\frac{1}{\sqrt{d}}z_{.,t'}^\top q\right).$$

It follows that

$$g'(q_a)\tilde{g}(q_a) = \frac{1}{\sqrt{d}}z_{at}\exp\left(\frac{1}{\sqrt{d}}z_{.,t}^\top q\right)\sum_{t'=1}^N \exp\left(\frac{1}{\sqrt{d}}z_{.,t'}^\top q\right)$$
$$= \frac{1}{\sqrt{d}}z_{at}\sum_{t'=1}^N \exp\left(\frac{1}{\sqrt{d}}\sum_{o=1}^d(z_{.,t} + z_{.,t'})^\top q\right)$$

and similarly

$$g(q_a)\tilde{g}'(q_a) = \exp\left(\frac{1}{\sqrt{d}}z_{.,t}^\top q\right)\sum_{t'=1}^N \frac{1}{\sqrt{d}}z_{at'}\exp\left(\frac{1}{\sqrt{d}}z_{.,t'}^\top q\right)$$
$$= \frac{1}{\sqrt{d}}\sum_{t'=1}^N z_{at'}\exp\left(\frac{1}{\sqrt{d}}\sum_{o=1}^d(z_{.,t} + z_{.,t'})^\top q\right).$$

Putting everything together, we obtain

$$\frac{\partial s_t}{\partial q_a} = \frac{1}{\sqrt{d}}\frac{\sum_{t'=1}^N(z_{at} - z_{at'})\exp\left(\frac{1}{\sqrt{d}}(z_{.,t} + z_{.,t'})^\top q\right)}{[\sum_{r=1}^N \exp\left(\frac{1}{\sqrt{d}}\sum_{o=1}^d z_{or}q_o\right)]^2}$$
$$= \frac{1}{\sqrt{d}}\sum_{t'=1}^N(z_{at} - z_{at'})\frac{\exp\left(\frac{1}{\sqrt{d}}z_{.,t}^\top q\right)}{\sum_{r=1}^N \exp\left(\frac{1}{\sqrt{d}}\sum_{o=1}^d z_{or}q_o\right)} \cdot \frac{\exp\left(\frac{1}{\sqrt{d}}z_{.,t'}^\top q\right)}{\sum_{r=1}^N \exp\left(\frac{1}{\sqrt{d}}\sum_{o=1}^d z_{or}q_o\right)}$$
$$= \frac{1}{\sqrt{d}}\sum_{t'=1}^N(z_{at} - z_{at'})\frac{\exp\left(\frac{1}{\sqrt{d}}\sum_{o=1}^d z_{ot}q_o\right)}{\sum_{r=1}^N \exp\left(\frac{1}{\sqrt{d}}\sum_{o=1}^d z_{or}q_o\right)} \cdot \frac{\exp\left(\frac{1}{\sqrt{d}}\sum_{o=1}^d z_{ot'}q_o\right)}{\sum_{r=1}^N \exp\left(\frac{1}{\sqrt{d}}\sum_{o=1}^d z_{or}q_o\right)}$$
$$= \frac{1}{\sqrt{d}}\sum_{t'=1}^N(z_{at} - z_{at'})\,\mathrm{softmax}(\frac{z^\top q}{\sqrt{d}})_t\,\mathrm{softmax}(\frac{z^\top q}{\sqrt{d}})_{t'}$$
$$= \frac{1}{\sqrt{d}}\sum_{t'=1}^N(z_{at} - z_{at'})s_t s_{t'}.$$

Using Eq. (12) leads to

$$
\begin{aligned}
\frac{\partial f_1(q_a)}{\partial q_a} = \frac{\partial \xi_l}{\partial q_a} &= \sum_{t=1}^{N} (Vz)_{lt} \frac{1}{\sqrt{d}} \sum_{t'=1}^{N} (z_{at} - z_{at'}) s_t s_{t'} \\
&= \frac{1}{\sqrt{d}} \sum_{tt'} (Vz)_{lt} z_{at} s_t s_{t'} - \frac{1}{\sqrt{d}} \sum_{tt'} (Vz)_{lt} z_{at'} s_t s_{t'} \\
&= \frac{1}{\sqrt{d}} \sum_{t=1}^{N} (Vz)_{lt} z_{at} s_t \underbrace{\sum_{t'=1}^{N} s_{t'}}_{=1} - \frac{1}{\sqrt{d}} \sum_{t=1}^{N} (Vz)_{lt} s_t \underbrace{\sum_{t=1}^{N} z_{at'} s_{t'}}_{=\mu(z_a)} \\
&= \frac{1}{\sqrt{d}} \sum_{t=1}^{N} (z_{at} - \mu(z_a)(Vz)_{lt} s_t \\
&= \sum_{t=1}^{N} \Delta_{at} (Vz)_{lt} s_t,
\end{aligned}
\tag{14}
$$

where $\Delta_{at} = \frac{z_{at} - \mu(z_a)}{\sqrt{d}}$ corresponds to the deviation from the mean embedding at dimension $a$, assuming the distribution probability follows the softmax $(s_t)_{t=1}^{N}$. We note that the $\Delta_{at}$ are the entries of

$$
\Delta_z = \frac{z}{\sqrt{d}} \left[ I_N - \text{softmax}(\frac{z^\top q}{\sqrt{d}}) \mathbb{1}_N^\top \right] \in \mathbb{R}^{d \times N}
$$

defined in Eq. (4). Hence, $\frac{\partial f_1(q_a)}{\partial q_a}$ can be understood as the $l$-element of a sentence embedding where the deviation from the mean embedding has rescaled the value matrix. We now proceed to the RHS of Eq. (11).

**RHS of Eq.** (11)**:** a simple application of the chain rule gives

$$
\frac{\partial f_2^i(q_a)}{\partial q_a} = \underbrace{\nabla f_5^i(q_a)}_{\in \mathbb{R}^{1 \times d}} \underbrace{\nabla f_4^i(\xi)}_{\in \mathbb{R}^{d \times d}} \underbrace{\nabla f_3^i(\bar{\xi})}_{\in \mathbb{R}^d} \underbrace{\sigma'(w_i^\top \bar{\xi})}_{\in \mathbb{R}},
$$

where we recall that $\bar{\xi} = \xi/\|\xi\|$. We have

$$
\nabla f_5^i(q_a) = \frac{\partial \xi}{\partial q_a} = \left[ \frac{\partial \xi_1}{\partial q_a}, \dots, \frac{\partial \xi_d}{\partial q_a} \right],
$$

and

$$
\nabla f_3^i(\bar{\xi}) = w_i.
$$

For $f_4^i$, we note that $f_4^i(\xi) = (h_1 \circ h_2)(\xi) \cdot \xi$, with $h_1 \colon \mathbb{R} \to \mathbb{R}, t \mapsto \frac{1}{t}$ and $h_2 \colon \mathbb{R}^d \to \mathbb{R}, x \mapsto \|x\|$. It leads to

$$
\begin{aligned}
\nabla f_4^i(\xi) &= \frac{\partial (h_1 \circ h2)(\xi)}{\partial \xi} \cdot \xi^\top + (h_1 \circ h_2)(\xi) \cdot \frac{\partial \xi}{\partial \xi} \\
&= \frac{\partial (h_1 \circ h2)(\xi)}{\partial \xi} \cdot \xi^\top + \frac{1}{\|\xi\|} I_d.
\end{aligned}
$$

Since we have $h_2(\xi) = \|\xi\| = \sqrt{\xi^\top \xi}$, we know that

$$
\frac{\partial h2(\xi)}{\partial \xi} = \frac{1}{2\sqrt{\xi^\top \xi}} \cdot 2\xi = \frac{\xi}{\|\xi\|}.
$$

It leads to

$$
\frac{\partial (h_1 \circ h2)(\xi)}{\partial \xi} = -\frac{1}{\|\xi\|^2} \cdot \frac{\xi}{\|\xi\|} = -\frac{\xi}{\|\xi\|^3}.
$$

Putting everything together, we obtain

$$\nabla f_4^i(\xi) = -\frac{\xi}{\|\xi\|^3}\xi^\top + \frac{1}{\|\xi\|}I_d = \frac{1}{\|\xi\|}\big(I_d - \bar{\xi}\bar{\xi}^\top\big).$$

Going back to $\frac{\partial f_2^i(q_a)}{\partial q_a}$, we have

$$\frac{\partial f_2^i(q_a)}{\partial q_a} = \left[\frac{\partial \xi_1}{\partial q_a}, \ldots, \frac{\partial \xi_d}{\partial q_a}\right]\frac{1}{\|\xi\|}\big(I_d - \bar{\xi}\bar{\xi}^\top\big)w_i\sigma'(w_i^\top \bar{\xi}) = \lambda M w_i \alpha_i,$$

where we introduce

$$\lambda = \left[\frac{\partial \xi_1}{\partial q_a}, \ldots, \frac{\partial \xi_d}{\partial q_a}\right] \in \mathbb{R}^{1\times d}, \quad M = \frac{1}{\|\xi\|}\big(I_d - \bar{\xi}\bar{\xi}^\top\big) \in \mathbb{R}^{d\times d}, \quad \alpha_i = \sigma'(w_i^\top \bar{\xi}), \tag{15}$$

to ease notations. Developing the matrix multiplication, we obtain

$$\begin{aligned}
\lambda M w_i \alpha_i &= \sum_{r=1}^{d}\frac{\partial \xi_r}{\partial q_a}[Mw_i]_r\alpha_i \\
&= \alpha_i\sum_{r=1}^{d}\sum_{t=1}^{N}\Delta_{at}(Vz)_{rt}s_t[Mw_i]_r &&\text{(using Eq. (14))} \\
&= \alpha_i\sum_{t=1}^{N}\Delta_{at}s_t\sum_{r=1}^{d}(Vz)_{rt}\sum_{o=1}^{d}M_{ro}(w_i)_o \\
&= \alpha_i\sum_{t=1}^{N}\Delta_{at}s_t\sum_{o=1}^{d}(w_i)_o\sum_{r=1}^{d}(Vz)_{rt}M_{ro} \\
&= \alpha_i\sum_{t=1}^{N}\Delta_{at}s_t\sum_{o=1}^{d}(w_i)_o\sum_{r=1}^{d}M_{ro}(Vz)_{rt} \\
&= \alpha_i\sum_{t=1}^{N}\Delta_{at}s_t\sum_{o=1}^{d}(w_i)_o\sum_{r=1}^{d}M_{or}^\top(Vz)_{rt} \\
&= \alpha_i\sum_{t=1}^{N}\Delta_{at}s_t\sum_{o=1}^{d}(w_i)_o\big[M^\top Vz\big]_{ot} \\
&= \alpha_i\sum_{t=1}^{N}\Delta_{at}s_t\sum_{o=1}^{d}(w_i)_o\tilde{M}_{ot},
\end{aligned}$$

where we introduce

$$\tilde{M} = M^\top Vz \in \mathbb{R}^{d\times N}. \tag{16}$$

Now multiplying by $(u_i)_l$ and summing over $i \in [h]$, we obtain

$$\begin{aligned}
\sum_{i=1}^{h}(u_i)_l\frac{\partial f_2^i(q_a)}{\partial q_a} &= \sum_{i=1}^{h}(u_i)_l\alpha_i\sum_{t=1}^{N}\Delta_{at}s_t\sum_{o=1}^{d}(w_i)_o\tilde{M}_{ot} \\
&= \sum_{t=1}^{N}\Delta_{at}s_t\sum_{o=1}^{d}\tilde{M}_{ot}\sum_{i=1}^{h}\alpha_i(u_i)_l(w_i)_o.
\end{aligned}$$

We can now put everything together to rewrite Eq. (11).

**LHS + RHS of Eq. (11):** We have

$$\frac{\partial \psi_l}{\partial q_a} = \frac{\partial \xi_l}{\partial q_a} + \sum_{t=1}^{N}\Delta_{at}s_t\sum_{o=1}^{d}\tilde{M}_{ot}\sum_{i=1}^{h}\alpha_i(u_i)_l(w_i)_o.$$

and Eq. (9) leads to

$$\frac{\partial \zeta_j}{\partial q_a} = \underbrace{\sum_{l=1}^{p} (W_U)_{jl} \frac{\partial \xi_l}{\partial q_a}}_{=\text{LHS}} + \underbrace{\sum_{l=1}^{p} (W_U)_{jl} \sum_{t=1}^{N} \Delta_{at} s_t \sum_{o=1}^{d} \tilde{M}_{ot} \sum_{i=1}^{h} \alpha_i (u_i)_l (w_i)_o}_{=\text{RHS}} .$$

(17)

The LHS can be rewritten as

$$
\begin{aligned}
\text{LHS} &= \sum_{l=1}^{p} (W_U)_{jl} \sum_{t=1}^{N} \Delta_{at} (Vz)_{lt} s_t \\
&= \sum_{t=1}^{N} \Delta_{at} s_t \sum_{l=1}^{p} (W_U)_{jl} (Vz)_{lt} \\
&= \sum_{t=1}^{N} \Delta_{at} s_t \sum_{l=1}^{p} E(j)_l (Vz)_{lt} \\
&= \sum_{t=1}^{N} \Delta_{at} s_t E(j)^\top (Vz)_{\cdot, t} \\
&= \left[ \sum_{t=1}^{N} \Delta_{at} s_t [(Vz)_{\cdot, t}]^\top \right] E(j).
\end{aligned}
$$

The RHS can be rewritten as

$$
\begin{aligned}
\text{RHS} &= \sum_{l=1}^{p} (W_U)_{jl} \sum_{t=1}^{N} \Delta_{at} s_t \sum_{o=1}^{d} \tilde{M}_{ot} \sum_{i=1}^{h} \alpha_i (u_i)_l (w_i)_o \\
&= \sum_{t=1}^{N} \Delta_{at} s_t \sum_{l=1}^{p} (W_U)_{jl} \sum_{i=1}^{h} \alpha_i (u_i)_l \sum_{o=1}^{d} \tilde{M}_{ot} (w_i)_o \\
&= \sum_{t=1}^{N} \Delta_{at} s_t \sum_{l=1}^{p} (W_U)_{jl} \sum_{i=1}^{h} \alpha_i (u_i)_l \sum_{o=1}^{d} (w_i)_o \tilde{M}_{ot} \\
&= \sum_{t=1}^{N} \Delta_{at} s_t \sum_{l=1}^{p} (W_U)_{jl} \sum_{i=1}^{h} \alpha_i (u_i)_l \sum_{o=1}^{d} W_{io} \tilde{M}_{ot} \\
&= \sum_{t=1}^{N} \Delta_{at} s_t \sum_{l=1}^{p} (W_U)_{jl} \sum_{i=1}^{h} \alpha_i (u_i)_l (W \tilde{M})_{it}.
\end{aligned}
$$

We now introduce the matrix

$$\tilde{P} = U \begin{pmatrix} \alpha_1 & 0 & \cdots \\ \cdots & \cdots & \cdots \\ \cdots & 0 & \alpha_h \end{pmatrix} W \tilde{M} \in \mathbb{R}^{d \times N}.$$

(18)

Its entry in row $l$ and column $t$ writes

$$
\tilde{P}_{lt} = \left[ U \begin{pmatrix} \alpha_1 & 0 & \cdots \\ \cdots & \cdots & \cdots \\ \cdots & 0 & \alpha_h \end{pmatrix} W \tilde{M} \right]_{lt}
$$

$$
= \sum_{i=1}^{h} \left[ U \begin{pmatrix} \alpha_1 & 0 & \cdots \\ \cdots & \cdots & \cdots \\ \cdots & 0 & \alpha_h \end{pmatrix} \right]_{li} (W\tilde{M})_{it}
$$

$$
= \sum_{i=1}^{h} \alpha_i U_{li} (W\tilde{M})_{it}
$$

$$
= \sum_{i=1}^{h} \alpha_i (u_i)_l (W\tilde{M})_{it}. \qquad\qquad \text{(as } U \text{ has columns } u_i)
$$

Plugging this term into the computation of the RHS leads to

$$
\text{RHS} = \sum_{t=1}^{N} \Delta_{at} s_t \sum_{l=1}^{p} (W_U)_{jl} \tilde{P}_{lt}
$$

$$
= \sum_{t=1}^{N} \Delta_{at} s_t \sum_{l=1}^{p} \tilde{P}_{lt} (W_U)_{jl}
$$

$$
= \sum_{t=1}^{N} \Delta_{at} s_t \sum_{l=1}^{p} \tilde{P}_{lt} E(j)_l \qquad\qquad \text{(as } W_U \text{ has rows } E(j)^\top)
$$

$$
= \sum_{t=1}^{N} \Delta_{at} s_t \sum_{l=1}^{p} \tilde{P}_{tl}^\top E(j)_l
$$

$$
= \sum_{t=1}^{N} \Delta_{at} s_t \tilde{P}_t^\top E(j).
$$

Going back to Eq. (17), we have

$$
\frac{\partial \zeta_j}{\partial q_a} = \left[ \sum_{t=1}^{N} \Delta_{at} s_t [(Vz)_{\cdot,t}]^\top \right] E(j) + \sum_{t=1}^{N} \Delta_{at} s_t \tilde{P}_t^\top E(j)
$$

$$
= \sum_{t=1}^{N} \Delta_{at} s_t \left[ (Vz)_{\cdot,t} + \tilde{P}_t^\top \right] E(j)
$$

$$
= \sum_{t=1}^{N} \Delta_{at} s_t \left[ (Vz)_{\cdot,t} + \tilde{P}_{\cdot,t} \right] E(j)
$$

$$
= \sum_{t=1}^{N} \Delta_{at} s_t \left[ Vz + \tilde{P} \right]_{\cdot,t} E(j)
$$

$$
= \sum_{t=1}^{N} \Delta_{at} s_t \left[ Vz + \tilde{P} \right]_t^\top E(j)
$$

$$
= \sum_{t=1}^{N} \Delta_{at} \left[ \begin{pmatrix} s_1 & 0 & \cdots \\ \cdots & \cdots & \cdots \\ \cdots & 0 & s_N \end{pmatrix} \left[ Vz + \tilde{P} \right]^\top \right]_t E(j)
$$

$$
= \Delta_a \begin{pmatrix} s_1 & 0 & \cdots \\ \cdots & \cdots & \cdots \\ \cdots & 0 & s_N \end{pmatrix} \left[ Vz + \tilde{P} \right]^\top E(j),
$$

where $\Delta_a$ is the $a$-th column of $\Delta$ defined in Eq. (4). Using Eq. (2) and Eq. (13), we notice that

$$A_z = \text{diag}(\text{softmax}(\frac{z^\top q}{\sqrt{d}})) = \text{diag}((s_t)_{t=1}^N) = \begin{pmatrix} s_1 & 0 & \cdots \\ \cdots & \cdots & \cdots \\ \cdots & 0 & s_N \end{pmatrix} \in \mathbb{R}^{N \times N}.$$

Using Eq. (3) and Eq. (15), we notice that

$$\Sigma_{\bar{\xi}} = \text{diag}(\sigma'(W\bar{\xi}) = \text{diag}((\sigma'(w_i^\top \bar{\xi})_{i=1}^h) = \text{diag}((\alpha_i)_{i=1}^h) = \begin{pmatrix} \alpha_1 & 0 & \cdots \\ \cdots & \cdots & \cdots \\ \cdots & 0 & \alpha_h \end{pmatrix} \in \mathbb{R}^{N \times N}.$$

Using Eq. (3) and Eq. (18), we thus have

$$\tilde{P} = U \begin{pmatrix} \alpha_1 & 0 & \cdots \\ \cdots & \cdots & \cdots \\ \cdots & 0 & \alpha_h \end{pmatrix} W\tilde{M} = U\Sigma_{\bar{\xi}}M\tilde{W}$$

and we obtain

$$\begin{aligned}
Vz + \tilde{P} &= Vz + U\Sigma_{\bar{\xi}}M\tilde{W} \\
&= Vz + U\Sigma_{\bar{\xi}}WM^\top Vz & \text{(using Eq. (16))} \\
&= Vz + U\Sigma_{\bar{\xi}}WMVz & (M \text{ symmetric from Eq. (15)}) \\
&= (I_d + U\Sigma_{\bar{\xi}}WM)Vz.
\end{aligned}$$

It follows that

$$\frac{\partial \zeta_j}{\partial q_a} = \Delta_a A_z \big[(I_d + U\Sigma_{\bar{\xi}}WM)Vz\big]^\top E(j).$$

It leads to $\nabla_q \zeta_j = \left(\frac{\partial \zeta_j}{\partial q_a}\right)_a \in \mathbb{R}^d$ that writes

$$\nabla_q \zeta_j = \Delta_z A_z \big[(I_d + U\Sigma_{\bar{\xi}}WM)Vz\big]^\top E(j).$$

The corresponding entries of $\nabla_\theta \zeta_j$ are the entries of $\nabla_q \zeta_j$. $\qquad\square$

**Attention value.** The following lemma provides the formula of $\nabla_V \xi_j$.

**Lemma 5** (Value). *Let $\Sigma_{\bar{\xi}} = \text{diag}(\sigma'(W\bar{\xi})) \in \mathbb{R}^{h \times h}$ following Eq. (3), $M = \frac{1}{\|\xi\|}\big(I_d - \bar{\xi}\bar{\xi}^\top\big) \in \mathbb{R}^{d \times d}$ following Eq. (15) and $Q = U\Sigma_{\bar{\xi}}WM \in \mathbb{R}^{d \times d}$. The entries of $\nabla_\theta \zeta_j$ associated to the value $V$ are the entries of $\nabla_U \zeta_j \in \mathbb{R}^{d \times d}$ flattened in $\mathbb{R}^{d^2}$ where we have*

$$\nabla_V \zeta_j = (I_d + Q^\top)E(j)z\,\text{softmax}(\frac{z^\top q}{\sqrt{d}}).$$

*Proof.* We want to compute the $\frac{\partial \zeta_j}{\partial \theta_n}$ for the $\theta_n$ that are entries of $V$. Using Eq. (9) we first need to derive $\frac{\partial \psi_l}{\partial \theta_n}$. As $\theta$ encompasses all the learnable weights, we will derive it depending on the value of $n$. Considering $n$ such that $\theta_n = V_{bc}$ for some $b, c \in [d]$ and using Eq. (10), we have

$$\frac{\partial \psi_l}{\partial \theta_n} = \frac{\partial \psi_l}{\partial V_{bc}} = \frac{\partial \xi_l}{\partial V_{bc}} + \frac{\partial}{\partial V_{bc}}(\sum_{i=1}^h (u_i)_l \sigma(w_i^\top \bar{\xi})) = \underbrace{\frac{\partial \beta_1(V_{bc})}{\partial V_{bc}}}_{\text{LHS}} + \underbrace{\sum_{i=1}^h (u_i)_l \frac{\partial \beta_2^i(V_{bc})}{\partial V_{bc}}}_{\text{RHS}}, \tag{19}$$

with

$$\begin{cases} \beta_1: & \mathbb{R} \to \mathbb{R}, & V_{bc} \mapsto \xi_l \\ \beta_2^i: & \mathbb{R} \to \mathbb{R}, & V_{bc} \mapsto (\sigma \circ \beta_3^i \circ \beta_4^i \circ \beta_5^i)(V_{bc}) = \sigma(w_i^\top \bar{\xi}) \\ \beta_3^i: & \mathbb{R}^d \to \mathbb{R}, & \bar{\xi} \mapsto w_i^\top \bar{\xi} \\ \beta_4^i: & \mathbb{R}^d \to \mathbb{R}^d, & \xi \mapsto \bar{\xi} = \frac{\xi}{\|\xi\|} \\ \beta_5^i: & \mathbb{R} \to \mathbb{R}^d, & V_{bc} \mapsto \xi. \end{cases}$$

We proceed in two steps and first compute the LHS of Eq. (19).

LHS of Eq. (19): we first note that

$$\xi_l = \sum_{t=1}^N (Vz)_{lt} s_t = \sum_{t=1}^N \sum_{c=1}^d V_{lc} z_{ct} s_t = \sum_{c=1}^d V_{lc} \sum_{t=1}^N z_{ct} s_t,$$

where $s_t = \mathrm{softmax}(\frac{z^\top q}{\sqrt{d}})_t$. We notice that $\xi_l$ depends only on the $(V_{lc})$, hence the gradients with respect to $V_{bc}$ for some $c \in [d]$ and $b \neq l$ equals zero. It follows that

$$\frac{\partial \beta_1(V_{bc})}{\partial V_{bc}} = \frac{\partial \xi_l}{\partial V_{bc}} = \begin{cases} \sum_{t=1}^N z_{ct} s_t & \text{if } b = l \\ 0 & \text{otherwise} \end{cases}. \tag{20}$$

**RHS of Eq.** (19)**:** a simple application of the chain rule gives

$$\frac{\partial \beta_2^i(V_{bc})}{\partial V_{bc}} = \underbrace{\nabla \beta_5^i(V_{bc})}_{\in \mathbb{R}^{1 \times d}} \underbrace{\nabla \beta_4^i(\xi)}_{\in \mathbb{R}^{d \times d}} \underbrace{\nabla \beta_3^i(\bar{\xi})}_{\in \mathbb{R}^d} \underbrace{\sigma'(w_i^\top \bar{\xi})}_{\in \mathbb{R}},$$

where we recall that $\bar{\xi} = \xi/\|\xi\|$. We have, using similar arguments as the computation of $\frac{\partial \xi_l}{\partial V_{bc}}$ in Eq. (20) that

$$\nabla \beta_5^i(V_{bc}) = \frac{\partial \xi}{\partial V_{bc}} = \left[ \frac{\partial \xi_1}{\partial V_{bc}}, \dots, \frac{\partial \xi_d}{\partial V_{bc}} \right] = \left[ 0, \dots, 0, \sum_{t=1}^N z_{ct} s_t, 0, \dots, 0 \right],$$

where the only non-zero entry is in position $b$ of the row-vector $\nabla \beta_5^i(V_{bc}) \in \mathbb{R}^{1 \times d}$. We notice that the functions $\beta_3^i$ and $\beta_4^i$ of Eq. (19) corresponds to the functions $f_3^i$ and $f_4^i$ of Eq. (11). Hence, the gradients are the same and we have

$$\nabla \beta_3^i(\bar{\xi}) = \nabla f_3^i(\bar{\xi}) = w_i$$

and

$$\nabla \beta_4^i(\bar{\xi}) = \nabla f_4^i(\bar{\xi}) = \frac{1}{\|\xi\|} \left( I_d - \bar{\xi} \bar{\xi}^\top \right).$$

Going back to $\frac{\partial \beta_2^i(V_{bc})}{\partial V_{bc}}$, we have

$$\frac{\partial \beta_2^i(V_{bc})}{\partial V_{bc}} = \left[ 0, \dots, 0, \sum_{t=1}^N z_{ct} s_t, 0, \dots, 0 \right] \frac{1}{\|\xi\|} \left( I_d - \bar{\xi} \bar{\xi}^\top \right) w_i \sigma'(w_i^\top \bar{\xi}) = \mu M w_i \alpha_i,$$

where we introduce

$$\mu = \left[ 0, \dots, 0, \sum_{t=1}^N z_{ct} s_t, 0, \dots, 0 \right] \in \mathbb{R}^{1 \times d}, \quad M = \frac{1}{\|\xi\|} \left( I_d - \bar{\xi} \bar{\xi}^\top \right) \in \mathbb{R}^{d \times d}, \quad \alpha_i = \sigma'(w_i^\top \bar{\xi}), \tag{21}$$

to ease notations, with $M, \alpha_i$ first defined in Eq. (15). Developing the matrix multiplication, we obtain

$$\mu M w_i \alpha_i = \alpha_i \sum_{t=1}^N z_{ct} s_t [M w_i]_b = \alpha_i \sum_{t=1}^N z_{ct} s_t \sum_{o=1}^d M_{bo}(w_i)_o.$$

It follows that

$$\sum_{i=1}^{h}(u_i)_l \frac{\partial \beta_2^i(V_{bc})}{\partial V_{bc}} = \sum_{i=1}^{h}(u_i)_l \alpha_i \sum_{t=1}^{N} z_{ct}s_t \sum_{o=1}^{d} M_{bo}(w_i)_o = \sum_{t=1}^{N} z_{ct}s_t \sum_{i=1}^{h} \alpha_i(u_i)_l \sum_{o=1}^{d} M_{bo}(w_i)_o.$$

We can now put everything together to rewrite Eq. (19).

LHS + RHS of Eq. (19): We have

$$\frac{\partial \psi_l}{\partial V_{bc}} = \frac{\partial \xi_l}{\partial V_{bc}} + \sum_{t=1}^{N} z_{ct}s_t \sum_{i=1}^{h} \alpha_i(u_i)_l \sum_{o=1}^{d} M_{bo}(w_i)_o$$

and Eq. (9) leads to

$$\frac{\partial \zeta_j}{\partial V_{bc}} = \underbrace{\sum_{l=1}^{p}(W_U)_{jl}\frac{\partial \xi_l}{\partial V_{bc}}}_{=\text{LHS}} + \underbrace{\sum_{l=1}^{p}(W_U)_{jl}\sum_{t=1}^{N} z_{ct}s_t \sum_{i=1}^{h} \alpha_i(u_i)_l \sum_{o=1}^{d} M_{bo}(w_i)_o}_{=\text{RHS}}. \tag{22}$$

Using Eq. (20), the LHS can be rewritten as

$$\text{LHS} = (W_U)_{jb}\sum_{t=1}^{N} z_{ct}s_t = E(j)_b \sum_{t=1}^{N} z_{ct}s_t.$$

The RHS can be rewritten as

$$\begin{aligned}
\text{RHS} &= \sum_{l=1}^{p}(W_U)_{jl}\sum_{t=1}^{N} z_{ct}s_t \sum_{i=1}^{h} \alpha_i(u_i)_l \sum_{o=1}^{d} M_{bo}(w_i)_o \\
&= \sum_{t=1}^{N} z_{ct}s_t \sum_{l=1}^{p}(W_U)_{jl}\left[\sum_{i=1}^{h} \alpha_i(u_i)_l \sum_{o=1}^{d} M_{bo}(w_i)_o\right] \\
&= \sum_{t=1}^{N} z_{ct}s_t \sum_{l=1}^{p}(W_U)_{jl}\left[\sum_{i=1}^{h} \alpha_i(u_i)_l \sum_{o=1}^{d} W_{io}M_{bo}\right] \qquad \text{(as } W \text{ has rows } w_i^\top) \\
&= \sum_{t=1}^{N} z_{ct}s_t \sum_{l=1}^{p}(W_U)_{jl}\left[\sum_{i=1}^{h} \alpha_i(u_i)_l(WM)_{ib}\right].
\end{aligned}$$

We now introduce the matrix

$$Q = U\begin{pmatrix} \alpha_1 & 0 & \cdots \\ \cdots & \cdots & \cdots \\ \cdots & 0 & \alpha_h \end{pmatrix} WM \in \mathbb{R}^{d\times d}. \tag{23}$$

Its entry in row $l$ and column $b$ writes

$$\begin{aligned}
Q_{lt} &= \left[U\begin{pmatrix} \alpha_1 & 0 & \cdots \\ \cdots & \cdots & \cdots \\ \cdots & 0 & \alpha_h \end{pmatrix} WM\right]_{lb} \\
&= \sum_{i=1}^{h}\left[U\begin{pmatrix} \alpha_1 & 0 & \cdots \\ \cdots & \cdots & \cdots \\ \cdots & 0 & \alpha_h \end{pmatrix}\right]_{li}(WM)_{ib} \\
&= \sum_{i=1}^{h} \alpha_i U_{li}(WM)_{ib} \\
&= \sum_{i=1}^{h} \alpha_i(u_i)_l(WM)_{ib}^\top. \qquad \text{(as } W \text{ has rows } w_i)
\end{aligned}$$

Plugging this term in the computation of the RHS leads to

$$
\begin{aligned}
\text{RHS} &= \sum_{t=1}^{N} z_{ct} s_t \sum_{l=1}^{p} (W_U)_{jl} Q_{lb} \\
&= \sum_{t=1}^{N} z_{ct} s_t \sum_{l=1}^{p} E(j)_l Q_{lb} \\
&= \sum_{t=1}^{N} z_{ct} s_t \sum_{l=1}^{p} Q_{bl}^{\top} E(j)_l \\
&= \sum_{t=1}^{N} z_{ct} s_t \left[ Q^{\top} E(j) \right]_b
\end{aligned}
$$

Going back to Eq. (22), we have

$$
\begin{aligned}
\frac{\partial \zeta_j}{\partial V_{bc}} &= E(j)_b \sum_{t=1}^{N} z_{ct} s_t + \sum_{t=1}^{N} z_{ct} s_t \left[ Q^{\top} E(j) \right]_b \\
&= \left[ E(j) + Q^{\top} E(j) \right]_b \sum_{t=1}^{N} z_{ct} s_t \\
&= \left[ (I_d + Q^{\top}) E(j) \right]_b \sum_{t=1}^{N} z_{ct} s_t.
\end{aligned}
$$

We introduce the vector $z \, \text{softmax}(\frac{z^{\top} q}{\sqrt{d}}) \in \mathbb{R}^d$ whose entries writes

$$
\left[ z \, \text{softmax}(\frac{z^{\top} q}{\sqrt{d}}) \right]_c = \sum_{t=1}^{N} z_{ct} \, \text{softmax}(\frac{z^{\top} q}{\sqrt{d}})_t = \sum_{t=1}^{N} z_{ct} s_t,
$$

where we used the fact that $s_t = \text{softmax}(\frac{z^{\top} q}{\sqrt{d}})_t$ following Eq. (13). It follows that

$$
\frac{\partial \zeta_j}{\partial V_{bc}} = \left[ (I_d + Q^{\top}) E(j) \right]_b \left[ z \, \text{softmax}(\frac{z^{\top} q}{\sqrt{d}}) \right]_c.
$$

Using Eq. (3) and Eq (23), we have

$$
Q = U \begin{pmatrix} \alpha_1 & 0 & \cdots \\ \cdots & \cdots & \cdots \\ \cdots & 0 & \alpha_h \end{pmatrix} W M = U \Sigma_{\bar{\xi}} W M.
$$

Hence, $\nabla_V \zeta_j = \left( \frac{\partial \zeta_j}{\partial V_{bc}} \right)_{b,c} \in \mathbb{R}^{d \times d}$ is a matrix equal to

$$
\nabla_V \zeta_j = \left( I_d + [U \Sigma_{\bar{\xi}} W M]^{\top} \right) E(j) \left[ z \, \text{softmax}(\frac{z^{\top} q}{\sqrt{d}}) \right]^{\top}.
$$

The corresponding entries of $\nabla_\theta \zeta_j$ amounts to the entries of $\nabla_V \zeta_j$ flattened in $\mathbb{R}^{d^2}$. This concludes the proof. $\qquad \square$

**Feed-forward receptors.** The following lemma provides the formula of $\nabla_W \xi_j$.

**Lemma 6** (Receptors). *Let $\Sigma_{\bar{\xi}} = \mathrm{diag}(\sigma'(W\bar{\xi})) \in \mathbb{R}^{h \times h}$ following Eq. (3). The entries of $\nabla_\theta \zeta_j$ associated to the assemblers $(w_i)_{i=1}^h$ are the entries of $\nabla_W \zeta_j \in \mathbb{R}^{h \times sd}$ flattened in $\mathbb{R}^{dh}$ where we have*

$$\nabla_W \zeta_j = \Sigma_{\bar{\xi}} U^\top E(j) \bar{\xi}^\top.$$

*Proof.* We want to compute the $\frac{\partial \zeta_j}{\partial \theta_n}$ for the $\theta_n$ that are entries of $W$. Using Eq. (9) we first need to derive $\frac{\partial \psi_l}{\partial \theta_n}$. As $\theta$ encompasses all the learnable weights, we will derive it depending on the value of $n$. Here, $\psi_l$ depends on all the $(w_i)_e$. Considering $n$ such that $\theta_n = (w_i)_e$ for some $i \in [h]$ and some $e \in [d]$ and using Eq. (10), we have

$$\frac{\partial \psi_l}{\partial \theta_n} = \frac{\partial \psi_l}{\partial (w_i)_e} = (u_i)_l \frac{\partial \sigma(w_i^\top \bar{\xi})}{\partial (w_i)_e}.$$

Noting that $w_i^\top \xi = \sum_{f=1}^d (w_i)_f \xi_f$ and using the chain rule, we have

$$\begin{aligned}
\frac{\partial \sigma(w_i^\top \xi / \|\xi\|)}{\partial (w_i)_e} &= \frac{\partial (w_i^\top \bar{\xi})}{\partial (w_i)_e} \sigma'(w_i^\top \bar{\xi}) \\
&= \frac{\partial (\sum_{f=1}^d (w_i)_f \bar{\xi}_f)}{\partial (w_i)_e} \sigma'(w_i^\top \bar{\xi}) \\
&= \frac{\partial ((w_i)_e \bar{\xi}_e)}{\partial (w_i)_e} \sigma'(w_i^\top \bar{\xi}) \\
&= \bar{\xi}_e \sigma'(w_i^\top \bar{\xi}).
\end{aligned}$$

We recall that $\sigma$ is the GeLU activation defined as $\sigma \colon x \to x\varphi(x)$ with $\varphi$ the cumulative distribution function of the standard normal distribution. By the fundamental theorem of analysis, it leads to

$$\sigma'(x) = \varphi(x) + x\varphi'(x) = \varphi(x) + xf(x),$$

with $f \colon u \to \frac{1}{\sqrt{2\pi}} \exp\left(-\frac{u^2}{2}\right)$ the probability density function of the standard normal distribution. Putting everything together, we have

$$\frac{\partial \psi_l}{\partial (w_i)_e} = (u_i)_l \bar{\xi}_e \sigma'(w_i^\top \bar{\xi}).$$

Using Eq. (9), we have

$$\begin{aligned}
\frac{\partial \zeta_j}{\partial (w_i)_e} &= \sum_{l=1}^p (W_U)_{jl} (u_i)_l \bar{\xi}_e \sigma'(w_i^\top \bar{\xi}) \\
&= \sum_{l=1}^p E(j)_l (u_i)_l \bar{\xi}_e \sigma'(w_i^\top \bar{\xi}) \\
&= E(j)^\top u_i \bar{\xi}_e \sigma'(w_i^\top \xi / \|\xi\|) \\
&= \sigma'(w_i^\top \bar{\xi}) u_i^\top E(j) \bar{\xi}_e.
\end{aligned}$$

To ease the notations, we denote $\alpha_i = \sigma'(w_i^\top \bar{\xi})$. Recalling that $\sigma$ and thus $\sigma'$ is entry-wise, it leads to $\nabla_W \zeta_j = (\frac{\partial \zeta_j}{\partial (w_i)_e})_{i,e} \in \mathbb{R}^{h \times d}$ that writes

$$
\begin{aligned}
\nabla_W \zeta_j &= \begin{pmatrix} \alpha_1 u_1^\top E(j) \\ \cdots \\ \alpha_h u_h^\top E(j) \end{pmatrix} \bar{\xi}^\top \\
&= \begin{pmatrix} \alpha_1 u_1^\top \\ \cdots \\ \alpha_h u_h^\top \end{pmatrix} E(j) \bar{\xi}^\top \\
&= \begin{pmatrix} \alpha_1 & 0 & \cdots \\ \cdots & \cdots & \cdots \\ \cdots & 0 & \alpha_h \end{pmatrix} U^\top E(j) \bar{\xi}^\top \\
&= \begin{pmatrix} \sigma'(w_1^\top \bar{\xi}) & 0 & \cdots \\ \cdots & \cdots & \cdots \\ \cdots & 0 & \sigma'(w_h^\top \bar{\xi}) \end{pmatrix} U^\top E(j) \bar{\xi}^\top \\
&= \mathrm{diag}(\sigma'(W\bar{\xi})) U^\top E(j) \bar{\xi}^\top,
\end{aligned}
$$

where $\mathrm{diag}(\sigma'(W\bar{\xi})) \in \mathbb{R}^{h \times h}$ is diagonal with entries $(\sigma'(w_i^\top \bar{\xi}))_{i=1}^h$. Using Eq. (3), it finally leads to

$$
\nabla_W \zeta_j = \Sigma_{\bar{\xi}} U^\top E(j) \bar{\xi}^\top,
$$

and the corresponding entries of $\nabla_\theta \zeta_j$ amounts to the entries of $\nabla_W \zeta_j$ flattened in $\mathbb{R}^{hd}$. This concludes the proof. $\square$

**Feed-forward assemblers.** The following lemma provides the formula of $\nabla_U \xi_j$.

> **Lemma 7** (Assemblers). *The entries of $\nabla_\theta \zeta_j$ associated to the assemblers $(u_i)_{i=1}^h$ are the entries of $\nabla_U \zeta_j \in \mathbb{R}^{d \times h}$ flattened in $\mathbb{R}^{dh}$ where we have*
>
> $$
> \nabla_U \zeta_j = E(j) \sigma(W\bar{\xi})^\top \in R^{d \times h}.
> $$

*Proof.* We want to compute the $\frac{\partial \zeta_j}{\partial \theta_n}$ for the $\theta_n$ that are entries of $U$. Using Eq. (9) we first need to derive $\frac{\partial \psi_l}{\partial \theta_n}$. As $\theta$ encompasses all the learnable weights, we will derive it depending on the value of $n$. As $\psi_l$ only depends on the $l$-elements of the $(u_i)$, we only consider indices $n$ such that $\theta_n = (u_i)_l$ for some $i \in [h]$ as the other values of $n$ will lead to a null gradient. Using Eq. (10), we have

$$
\frac{\partial \psi_l}{\partial \theta_n} = \frac{\partial \psi_l}{\partial (u_i)_l} = \frac{\partial (u_i)_l}{\partial (u_i)_l} \sigma(w_i^\top \bar{\xi}) = \sigma(w_i^\top \bar{\xi}).
$$

Using Eq. (9), we have

$$
\frac{\partial \zeta_j}{\partial (u_i)_l} = (W_U)_{jl} \sigma(w_i^\top \bar{\xi}) = E(j)_l \sigma(w_i^\top \bar{\xi}).
$$

Hence, recalling that $\sigma$ is applied element-wise, $\nabla_U \zeta_j = (\frac{\partial \zeta_j}{\partial (u_i)_l})_{l,i} \in \mathbb{R}^{d \times h}$ is a matrix equal to

$$
\nabla_U \zeta_j = E(j) \sigma(W\bar{\xi})^\top,
$$

and the corresponding entries of $\nabla_\theta \zeta_j$ amounts to the entries of $\nabla_U \zeta_j$ flattened in $\mathbb{R}^{dh}$. This concludes the proof. $\square$

We can now put everything together. Using Lemma 3 in combination with the gradientsderived in Lemma 4, Lemma 5, Lemma 6 and Lemma 7 concludes the proof of Lemma 2. $\square$

## C.2  Proof of Proposition 1

We detail below the proof of Proposition 3.

*Proof.* We assume the gradient descent continues until convergence at a critical point and discard other practical considerations such as early stopping. Following Chi et al. (2019b) and using Definition 1, we know that the parameters $\theta$ of the transformer verify

$$\nabla_\theta \mathcal{L}(\theta) = 0 \tag{24}$$

From Lemma 2, we know that $\nabla_\theta \mathcal{L}(\theta) \in \mathbb{R}^m$ writes

$$\nabla_\theta \mathcal{L}(\theta) = [\nabla_q \mathcal{L}(\theta), \nabla_V \mathcal{L}(\theta), \nabla_W \mathcal{L}(\theta), \nabla_U \mathcal{L}(\theta)], \tag{25}$$

using $[\cdot]$ as the vector concatenation and flattening matrices whenever needed. We now reformulate the gradients of each weight group. We have

$$\nabla_q \mathcal{L}(\theta) = \sum_{j=1}^{p} \mathbb{E}_x[(\hat{p}_\theta(y=j|x) - p(y=j|x)) \cdot \nabla_q \zeta_j]$$

$$= \sum_{j=1}^{p} \mathbb{E}_x\left[(\hat{p}_\theta(y=j|x) - p(y=j|x)) \cdot \left(\Delta_z A_z (Vz)^\top (I_d + Q^\top) E(j)\right)\right]$$

$$= \mathbb{E}_x\left[\sum_{j=1}^{p} (\hat{p}_\theta(y=j|x) - p(y=j|x)) \cdot \left(\Delta_z A_z (Vz)^\top (I_d + Q^\top) E(j)\right)\right]$$

(by linearity of the expectation)

$$= \mathbb{E}_x\left[\Delta_z A_z (Vz)^\top (I_d + Q^\top) \cdot \sum_{j=1}^{p} (\hat{p}_\theta(y=j|x) - p(y=j|x)) E(j)\right] \quad (\hat{p}_\theta(y=j|x) - p(y=j|x) \in \mathbb{R})$$

$$= \mathbb{E}_x\left[\Delta_z A_z (Vz)^\top (I_d + Q^\top) \mathscr{C}_x\right]. \quad \text{(by definition of } \mathscr{C}_x\text{)}$$

The same arguments lead to

$$\nabla_V \mathcal{L}(\theta) = \sum_{j=1}^{p} \mathbb{E}_x[(\hat{p}_\theta(y=j|x) - p(y=j|x)) \cdot \nabla_V \zeta_j]$$

$$= \sum_{j=1}^{p} \mathbb{E}_x\left[(\hat{p}_\theta(y=j|x) - p(y=j|x)) \cdot \left((I_d + Q^\top) E(j)\left[z\, \mathrm{softmax}(\frac{z^\top q}{\sqrt{d}})\right]^\top\right)\right]$$

$$= \mathbb{E}_x\left[\sum_{j=1}^{p} (\hat{p}_\theta(y=j|x) - p(y=j|x)) \cdot \left((I_d + Q^\top) E(j)\left[z\, \mathrm{softmax}(\frac{z^\top q}{\sqrt{d}})\right]^\top\right)\right]$$

$$= \mathbb{E}_x\left[(I_d + Q^\top) \cdot \left(\sum_{j=1}^{p} (\hat{p}_\theta(y=j|x) - p(y=j|x)) E(j)\right) \cdot \left[z\, \mathrm{softmax}(\frac{z^\top q}{\sqrt{d}})\right]^\top\right]$$

$$= \mathbb{E}_x\left[(I_d + Q^\top) \mathscr{C}_x \left[z\, \mathrm{softmax}(\frac{z^\top q}{\sqrt{d}})\right]^\top\right],$$

then to

$$\nabla_W \mathcal{L}(\theta) = \sum_{j=1}^{p} \mathbb{E}_x [(\hat{p}_\theta(y = j|x) - p(y = j|x)) \cdot \nabla_W \zeta_j]$$

$$= \sum_{j=1}^{p} \mathbb{E}_x \left[ (\hat{p}_\theta(y = j|x) - p(y = j|x)) \cdot \left( \Sigma_{\bar{\xi}} U^\top E(j) \bar{\xi}^\top \right) \right]$$

$$= \mathbb{E}_x \left[ \sum_{j=1}^{p} (\hat{p}_\theta(y = j|x) - p(y = j|x)) \cdot \left( \Sigma_{\bar{\xi}} U^\top E(j) \bar{\xi}^\top \right) \right]$$

$$= \mathbb{E}_x \left[ \Sigma_{\bar{\xi}} U^\top \cdot \left( \sum_{j=1}^{p} (\hat{p}_\theta(y = j|x) - p(y = j|x)) E(j) \right) \cdot \bar{\xi}^\top \right]$$

$$= \mathbb{E}_x \left[ \Sigma_{\bar{\xi}} U^\top \mathscr{C}_x \bar{\xi}^\top \right],$$

and finally to

$$\nabla_U \mathcal{L}(\theta) = \sum_{j=1}^{p} \mathbb{E}_x [(\hat{p}_\theta(y = j|x) - p(y = j|x)) \cdot \nabla_U \zeta_j]$$

$$= \sum_{j=1}^{p} \mathbb{E}_x \left[ (\hat{p}_\theta(y = j|x) - p(y = j|x)) \cdot \left( E(j) \sigma(W\bar{\xi})^\top \right) \right]$$

$$= \mathbb{E}_x \left[ \sum_{j=1}^{p} (\hat{p}_\theta(y = j|x) - p(y = j|x)) \cdot \left( E(j) \sigma(W\bar{\xi})^\top \right) \right]$$

$$= \mathbb{E}_x \left[ \left( \sum_{j=1}^{p} (\hat{p}_\theta(y = j|x) - p(y = j|x)) E(j) \right) \cdot \sigma(W\bar{\xi})^\top \right]$$

$$= \mathbb{E}_x \left[ \mathscr{C}_x \sigma(W\bar{\xi})^\top \right].$$

The equality to 0 of the vector in $\mathbb{R}^m$ is equivalent to each of its components being equal to 0, and in particular, this is the case of the gradients with respect to $q, V, W,$ and $U$. Combining Eq. (24) with Lemma 2 leads to the following stationarity conditions

$$\begin{cases} \mathbb{E}_x \left[ \Delta_z A_z (Vz)^\top (I_d + Q^\top) \mathscr{C}_x \right] = 0 \\ \mathbb{E}_x \left[ (I_d + Q^\top) \mathscr{C}_x \left[ z \operatorname{softmax}(\frac{z^\top q}{\sqrt{d}}) \right]^\top \right] = 0 \\ \mathbb{E}_x \left[ \Sigma_{\bar{\xi}} U^\top \mathscr{C}_x \bar{\xi}^\top \right] = 0 \\ \mathbb{E}_x \left[ \mathscr{C}_x \sigma(W\bar{\xi})^\top \right] = 0. \end{cases}$$

Once those conditions are established, the rest of the proof is straightforward. Indeed, it suffices to recall that the clustering heads described in Section 3 capture all the invariants of the problem. This leads to a perfect separation of the clusters that can be seen in Fig. 1. In particular, the transformer correctly classifies all the sequences, which amounts to the $\hat{p}_\theta(y = j|x)$ and the $p(y = j|x)$ being aligned. By definition, this implies $\mathscr{C}_x = 0$ for all $x$, and noticing the linear dependency of the gradients with $\mathscr{C}_x$ ensures that the stationary conditions previously stated are verified. Hence, a transformer that implements a circuit head has parameters $\theta$ for which $\nabla \mathcal{L}(\theta) = 0$ holds and that can thus be reached via gradient descent. This implies circuit heads can emerge during the training of the transformer. This concludes the proof. $\square$

### C.3 Proof of Proposition 2

We detail below the proof of Proposition 2 whose formal statement is given in Proposition 4.

*Proof.* We first decompose the gradient of the loss using Eq. (25). Gathering the terms by group leads to

$$\|\nabla_\theta \mathcal{L}(\theta)\|^2 = \sum_{n=1}^m \left| \frac{\partial \mathcal{L}(\theta)}{\partial \theta_n} \right|^2 = \|\nabla_q \mathcal{L}(\theta)\|^2 + \|\nabla_V \mathcal{L}(\theta)\|^2 + \|\nabla_W \mathcal{L}(\theta)\|^2 + \|\nabla_U \mathcal{L}(\theta)\|^2, \tag{26}$$

where the norm is the Euclidean one with matrices flattened as vectors. Similarly to the proof of Proposition 3, we have

$$\begin{cases} \nabla_q \mathcal{L}(\theta) = \mathbb{E}_x \big[ \Delta_z A_z (Vz)^\top (I_d + Q^\top) \mathscr{C}_x \big] \\ \nabla_V \mathcal{L}(\theta) = \mathbb{E}_x \Big[ (I_d + Q^\top) \mathscr{C}_x \big[ z \operatorname{softmax}(\frac{z^\top q}{\sqrt{d}}) \big]^\top \Big] \\ \nabla_W \mathcal{L}(\theta) = \mathbb{E}_x \big[ \Sigma_{\bar{\xi}} U^\top \mathscr{C}_x \bar{\xi}^\top \big] \\ \nabla_U \mathcal{L}(\theta) = \mathbb{E}_x \big[ \mathscr{C}_x \sigma(W\bar{\xi})^\top \big]. \end{cases}$$

We now proceed with upper-bounding each term of Eq. (26). For clarity of the proofs, we first recall the following basic relations.

---

**Lemma 8** (Properties). *Let $a \in \mathbb{R}^m, b \in \mathbb{R}^n, A \in \mathbb{R}^{n \times m}$ and $B \in \mathbb{R}^{m \times l}$. We have*

1. $\|ab^\top\| = \|a\| \cdot \|b\|$.
2. $\|Ab\| \le \|A\|_{\mathrm{op}} \|b\|$.
3. *The operator norm (resp. the Frobenius norm) is sub-multiplicative, i.e., $\|AB\|_{\mathrm{op}} \le \|A\|_{\mathrm{op}} \|B\|_{\mathrm{op}}$.*
4. *If $A$ has elements bounded by $C > 0$, then $\|A\|_{\mathrm{op}} \le C\sqrt{nm}$.*
5. *If $P$ be an orthogonal projection, then, we have $\|P\|_{\mathrm{op}} = 1$.*
6. *The singular values of $A$ and $A^\top$ coincides. In particular, $\|A\|_{\mathrm{op}} = \|A^\top\|_{\mathrm{op}}$.*

---

*Proof.* The first point follows from

$$\|ab^\top\|^2 = \sum_{i,j} (ab^\top)_{ij}^2 = \sum_{i,j} (a_i b_j)^2 = \left( \sum_i a_i^2 \right) \cdot \left( \sum_j b_j^2 \right) = \|a\|^2 \cdot \|b\|^2,$$

where taking the square root gives the desired equality.

The second point comes from the definition of the operator norm

$$\|A\|_{\mathrm{op}} = \sup\{ \frac{\|Ax\|}{\|x\|} : x \in \mathbb{R}^m, x \ne 0 \} = \sup\{ \|Ax\| : x \in \mathbb{R}^m, \|x\| = 1 \}.$$

It implies that for any $x \in \mathbb{R}^m$ non-zero, by definition of the sup, we have

$$\|Ax\| = \frac{\|Ax\|}{\|x\|} \cdot \|x\| \le \|A\|_{\mathrm{op}} \|x\|.$$

The inequality holds (with equality) for $x = 0$.

The third point is simply a property of matrix norms.

The fourth point comes from the definition of operator norm. Let $x \in \mathbb{R}^m$ with unit norm. We have

$$\|Ax\|^2 = \sum_{i=1}^n \left( \sum_{j=1}^m A_{ij} x_j \right)^2 \le \sum_{i=1}^n \left( \sum_{j=1}^m A_{ij}^2 \right) \left( \sum_{j=1}^m x_j^2 \right) \le nm \cdot C^2, \tag{27}$$

using the fact that $\|x\|^2 = \sum_{j=1}^m x_j^2 = 1$ and the bound on the entries of $A$ for the last inequality. Taking the root square and the sup on the left-hand term gives the desired formula.

For the fifth point, we recall that an orthogonal projection is symmetric. Hence, we have $P = P^2 = p^\top P$, which also ensures $P$ to be semi-definite positive, i.e., with nonnegative eigenvalues. Let $\lambda$ be such an eigenvalue and $v$ an eigenvector. We must have $Pv = \lambda v$ but applying $P$ on this relation also leads to

$$Pv = P^2 v = P(Pv) = P(\lambda v) = \lambda^2 v.$$

Taking the norm forces $\lambda = \lambda^2$ which implies $\lambda \in \{0, 1\}$. As the singular values of $P$ are its eigenvalues (by the symmetry of $P$), it implies that $\|P\|_{\mathrm{op}} \leq 1$. Moreover, writing the singular value decomposition of $P$ leads to $P = \sum_i \sigma_i u_i v_i^\top$. Let $u_k$ be in the range of $P$ (it suffices to show that $Pv_k = \sigma_k u_k$ and since $P \neq 0$, there is at least one $\sigma_k$ non-zero which provides at least one $u_k$ in the range of $P$). Using $P^2 = P$ implies that

$$Pu_k = u_k \iff \sum_i \sigma_i u_i v_i^\top u_k = u_k \iff \sum_i (\sigma_i v_i^\top u_k) u_i = u_k.$$

Since the $(u_i)$ forms an orthogonal basis, it forces $(\sigma_k v_k^\top u_k) = 1$. A simple application of Cauchy-Schwartz (recalling that the $u_i, v_i$ have unit norm) leads to $v_k^\top u_k \leq 1$, and finally it implies that $\sigma_k \geq 1$. This forces by definition of the operator norm $\|P\|_{\mathrm{op}} \geq 1$. Putting everything together gives $\|P\|_{\mathrm{op}} = 1$.

For the last point, assume $A$ has a singular value decomposition that writes

$$A = \sum_{i=1}^{\min\{n,m\}} \sigma_i u_i v_i^\top,$$

where the $u_i \in \mathbb{R}^n, v_i \in \mathbb{R}^m$ are orthogonal basis. Hence, we have

$$A^\top = \sum_{i=1}^{\min\{n,m\}} \sigma_i v_i u_i^\top,$$

which shows that the singular values are the same. Since the operator norm computes the largest singular value, we have the last equality. $\qquad\square$

In the rest, we upper-bound all the terms by extensively using the monotonicity of the expectation.

**Query term.** We have

$$\begin{aligned}
\|\nabla_q \mathcal{L}(\theta)\|^2 &= \|\mathbb{E}_x[\Delta_z A_z (Vz)^\top (I_d + Q^\top) \mathscr{C}_x](\theta)\|^2 \\
&\leq \mathbb{E}_x[\|\Delta_z A_z (Vz)^\top (I_d + Q^\top) \mathscr{C}_x\|^2] && \text{(Jensen inequality since } \|\cdot\|^2 \text{ is convex)} \\
&\leq \mathbb{E}_x[\|\Delta_z A_z (Vz)^\top (I_d + Q^\top)\|_{\mathrm{op}}^2 \|\mathscr{C}_x\|^2] && \text{(Point (2) of Lemma 8)}
\end{aligned}$$

Using the fact that operator norms are sub-multiplicative, we have

$$\|\Delta_z A_z (Vz)^\top (I_d + Q^\top)\|_{\mathrm{op}} \leq \|\Delta_z\|_{\mathrm{op}} \|A_z\|_{\mathrm{op}} \|(Vz)^\top\|_{\mathrm{op}} \|(I_d + Q^\top)\|_{\mathrm{op}}.$$

Recalling from Eq. (4) that

$$\Delta_z = \frac{z}{\sqrt{d}}\left[I_N - \mathrm{softmax}(\frac{z^\top q}{\sqrt{d}}) \mathbb{1}_N^\top\right],$$

and using the fact that the embeddings $z \in \mathbb{R}^{d \times M}$ have columns in $\mathbb{S}^d$ (thanks to the RMS-norm), we have that $I_N - \mathrm{softmax}(\frac{z^\top q}{\sqrt{d}}) \mathbb{1}_N^\top$ has entries bounded by 2. Indeed, given that

$$\forall t \in [N], \|z_{\cdot,t}\|^2 = \sum_{i=1}^d z_{dt}^2 = 1,$$

we know that the $z_{dt}$ are bounded by one. As the softmax outputs a probability vector, the $z_{dt}$ are reweighted by elements in $[0, 1]$. This implies that the entries of $\text{softmax}(\frac{z^\top q}{\sqrt{d}})\mathbb{1}_N^\top$ (a matrix with rows equal to $\text{softmax}(\frac{z^\top q}{\sqrt{d}})$) are bounded by 1. Taking the minus sign into account leads to $I_N - \text{softmax}(\frac{z^\top q}{\sqrt{d}})\mathbb{1}_N^\top$ with entries bounded by 2 and using the point (3) and (4) of Lemma 8, we have

$$\|\Delta_z\|_{\text{op}} \leq \frac{1}{\sqrt{d}}\|z\|_{\text{op}}\|I_N - \text{softmax}(\frac{z^\top q}{\sqrt{d}})\mathbb{1}_N^\top\|_{\text{op}}$$

$$\leq \frac{1}{\sqrt{d}}\|z^\top\|_{\text{op}} \cdot (2N) \qquad\qquad \text{(last point of Lemma 8)}$$

Moreover, using a proof similar to the point (4) of Lemma 8 (in particular involving Eq. (27)), one can show that for any $A \in \mathbb{R}^{n \times m}$,

$$\|A\|_{\text{op}} \leq \sqrt{m}\max_{i \in [m]}\|A_i\|.$$

Applying the previous inequality to $z^\top \in \mathbb{R}^{N \times d}$ that has rows $z_{.,t}$ with unit norm leads to

$$\|z^\top\|_{\text{op}} \leq \sqrt{d}. \tag{28}$$

We thus obtain $\|\Delta_z\|_{\text{op}} \leq 2N$. Recalling from (2) that

$$A_z = \text{diag}\left(\text{softmax}(\frac{z^\top q}{\sqrt{d}})\right)$$

is diagonal with entries that form a probability vector, we know that

$$\|A_z\|_{\text{op}} = \sigma_{\max}(A_z) = \lambda_{\max}(A_z) = \max_{t \in [N]}\{\text{softmax}(\frac{z^\top q}{\sqrt{d}})_t\} \leq 1.$$

Then, using Eq. (28), we notice that

$$\|(Vz)^\top\|_{\text{op}} = \|z^\top V^\top\|_{\text{op}} \leq \|z^\top\|_{\text{op}}\|V^\top\|_{\text{op}} \leq \sqrt{d}\|V^\top\|_{\text{op}} = \sqrt{d}\|V\|_{\text{op}},$$

where the last equality comes from the last point of Lemma 8.

We now proceed to $I + Q^\top$. Using the triangular inequality leads to $\|I + Q^\top\|_{\text{op}} \leq \|I\|_{\text{op}} + \|Q^\top\|_{\text{op}}$. We know that $\|I\|_{\text{op}} = 1$. Moreover, we have

$$\|Q^\top\|_{\text{op}} = \|M(U\Sigma_{\bar{\xi}}W)^\top\|_{\text{op}} \leq \|M\|_{\text{op}} \cdot \|(U\Sigma_{\bar{\xi}}W)^\top\|_{\text{op}}.$$

Using the fact that $M$ is an orthogonal projection ensures from the point (6) of Lemma 8 that $\|M\|_{\text{op}} = 1$. Moreover, using the points (3) and (6) of Lemma 8 leads to

$$\|(U\Sigma_{\bar{\xi}}W)^\top\|_{\text{op}} = \|U\Sigma_{\bar{\xi}}W\|_{\text{op}} \leq \|U\|_{\text{op}}\|\Sigma_{\bar{\xi}}\|_{\text{op}}\|W\|_{\text{op}}.$$

Noting that $\Sigma_{\bar{\xi}}$ is diagonal with entries $\sigma'(w_i^\top\bar{\xi})$ (Eq. (3)) and recalling that $\sigma'(x) = \psi(x) + xf(x)$ with $\varphi$ the cumulative distribution function of the normal distribution and $f$ its probability density function (in particular, $\varphi$ has values in $[0, 1]$ and $f$ in $[0, 1/\sqrt{2\pi}]$), we know that $|\sigma'(x)| \leq 1 + (1/\sqrt{2\pi})|x|$. Since $\sigma'$ is applied entry-wise, using again the point (2) of Lemma 8 leads to

$$\|\Sigma_{\bar{\xi}}\|_{\text{op}} \leq 1 + \frac{1}{\sqrt{2\pi}}\|W\bar{\xi}\| \leq 1 + \frac{1}{\sqrt{2\pi}}\|W\|_{\text{op}}.$$

Putting everything together leads to

$$\|\Delta_z A_z (Vz)^\top (I_d + Q^\top)\|_{\text{op}} \leq 2n\sqrt{d}\|V\|_{\text{op}}(1 + \sqrt{\frac{2}{\pi}}\|W\|_{\text{op}}^2\|U\|_{\text{op}}).$$

Since the weight matrices are randomly initialized and learned by gradient descent whose updates have the form $\mathbb{E}_x[\varphi(\text{parameters})]$, they do not depend on the expectation with respect to the training data distribution. We can thus take those terms outside of the expectation which leads to

$$\|\nabla_q\mathcal{L}(\theta)\|^2 \leq \left[2n\sqrt{d}\|V\|_{\text{op}}(1 + \sqrt{\frac{2}{\pi}}\|W\|_{\text{op}}^2\|U\|_{\text{op}})\right]^2 \cdot \mathbb{E}_x[\|\mathscr{C}_x\|^2]. \tag{29}$$

**Value term.** We have

$$\|\nabla_V \mathcal{L}(\theta)\|^2 \le \mathbb{E}_x\left[\|\underbrace{(I_d + Q^\top)\mathscr{C}_x}_{\in \mathbb{R}^d}\underbrace{\left[z\,\mathrm{softmax}(\frac{z^\top q}{\sqrt{d}})\right]^\top}_{\in \mathbb{R}^{1\times d}}\|^2\right] \qquad \text{(Jensen inequality)}$$

$$\le \mathbb{E}_x\left[\|(I_d + Q^\top)\mathscr{C}_x\|^2\left\|\left[z\,\mathrm{softmax}(\frac{z^\top q}{\sqrt{d}})\right]^\top\right\|^2\right] \qquad \text{(Point (1) of Lemma 8)}$$

Again, using point (2) of Lemma 8 and the derivations done for the query term, we have

$$\|(I_d + Q^\top)\mathscr{C}_x\| \le \|(I_d + Q^\top)\|_{\mathrm{op}} \cdot \|\mathscr{C}_x\| \le (1 + 2\|W\|_{\mathrm{op}}\|U\|_{\mathrm{op}}) \cdot \|\mathscr{C}_x\|.$$

Similarly, we have

$$\left\|\left[z\,\mathrm{softmax}(\frac{z^\top q}{\sqrt{d}})\right]^\top\right\| = \left\|z\,\mathrm{softmax}(\frac{z^\top q}{\sqrt{d}})\right\| \le \|z\|_{\mathrm{op}}\|\mathrm{softmax}(\frac{z^\top q}{\sqrt{d}})\|.$$

As shown in the proof of the query term, we have $\|z\|_{\mathrm{op}} = \|z^\top\|_{\mathrm{op}} \le \sqrt{d}$ and $p = \mathrm{softmax}(\frac{z^\top q}{\sqrt{d}})$ is a probability vector which implies

$$\|p\|^2 = \sum_{i=1}^N p_i^2 \le \sum_{i=1}^N p_i = 1,$$

where the first inequality comes from $x \mapsto x^2 \le x \mapsto x$ on $[0,1]$. It follows that

$$\left\|\left[z\,\mathrm{softmax}(\frac{z^\top q}{\sqrt{d}})\right]^\top\right\| \le \sqrt{d}.$$

In summary, with the same arguments as before, we can take the weight terms outside of the expectation and we obtain

$$\|\nabla_V \mathcal{L}(\theta)\|^2 \le d(1 + 2\|W\|_{\mathrm{op}}\|U\|_{\mathrm{op}})^2 \cdot \mathbb{E}_x[\|\mathscr{C}_x\|^2]. \tag{30}$$

**Receptors terms.** With similar arguments, we have

$$\|\nabla_W \mathcal{L}(\theta)\|^2 = \|\mathbb{E}_x[\Sigma_{\bar{\xi}} U^\top \mathscr{C}_x \bar{\xi}^\top]\|^2$$

$$\le \mathbb{E}_x\left[\|\underbrace{\Sigma_{\bar{\xi}} U^\top \mathscr{C}_x}_{\in \mathbb{R}^d}\underbrace{\bar{\xi}^\top}_{\in \mathbb{R}^{1\times d}}\|^2\right] \qquad \text{(Jensen)}$$

$$\le \mathbb{E}_x[\|\Sigma_{\bar{\xi}} U^\top \mathscr{C}_x\|^2\|\bar{\xi}^\top\|^2] \qquad \text{(Point (1) of Lemma 8)}$$

By definition, $\bar{\xi} = \xi/\|\xi\|$ has unit norm. Moreover, using the point (2) of Lemma 8, we know that

$$\|\Sigma_{\bar{\xi}} U^\top \mathscr{C}_x\| \le \|\Sigma_{\bar{\xi}} U^\top\|_{\mathrm{op}} \cdot \|\mathscr{C}_x\| \le \|\Sigma_{\bar{\xi}}\|_{\mathrm{op}} \cdot \|U^\top\|_{\mathrm{op}} \cdot \|\mathscr{C}_x\|.$$

where the last inequality comes from the sub-multiplicity of the operator norm. As we showed in the deviation of the query term, we have $\|\Sigma_{\bar{\xi}}\|_{\mathrm{op}} \le 2$ and we know that $\|U^\top\|_{\mathrm{op}} = \|U\|_{\mathrm{op}}$. In summary, taking the weight terms outside of the expectation, we have

$$\|\nabla_W \mathcal{L}(\theta)\|^2 \le 4\|U\|_{\mathrm{op}}^2 \mathbb{E}_x[\|\mathscr{C}_x\|^2]. \tag{31}$$

**Assemblers terms.** The same arguments as before lead to

$$\|\nabla_U \mathcal{L}(\theta)\|^2 = \|\mathbb{E}_x[\mathscr{C}_x \sigma(W\bar{\xi})^\top]\|^2 \leq \mathbb{E}_x[\|\mathscr{C}_x \sigma(W\bar{\xi})^\top\|^2] \leq \mathbb{E}_x[\|\mathscr{C}_x\|^2 \cdot \|\sigma(W\bar{\xi})^\top\|^2]$$

where the last inequality comes from the first point of Lemma 8. Recalling that the GeLU function is defined as $x \leq x\varphi(x)$ with $\varphi$ the cumulative distribution function of the standard normal distribution (in particular, $\varphi$ outputs values in $[0, 1]$), we know that for any real number $x$, $|\sigma(x)| \leq |x|$. Using again the point (2) of Lemma 8 and that $\bar{\xi}$ has unit norm, since $\sigma$ is applied entry-wise, we have

$$\|\sigma(W\bar{\xi})^\top\| = \|\sigma(W\bar{\xi})\| \leq \|W\bar{\xi}\| \leq \|W\|_{\mathrm{op}}.$$

In summary, taking the weight terms outside of the expectation, we have

$$\|\nabla_U \mathcal{L}(\theta)\|^2 \leq \|W\|_{\mathrm{op}}^2 \mathbb{E}_x[\|\mathscr{C}_x\|^2]. \tag{32}$$

**Conclusion.** Putting (29), (30), (31) and (32) together and using (26) leads to

$$\|\nabla_\theta \mathcal{L}(\theta)\|^2$$
$$\leq \left[\left[2n\sqrt{d}\|V\|_{\mathrm{op}}(1 + \sqrt{\frac{2}{\pi}}\|W\|_{\mathrm{op}}^2\|U\|_{\mathrm{op}})\right]^2 + d(1 + 2\|W\|_{\mathrm{op}}\|U\|_{\mathrm{op}})^2 + 4\|U\|_{\mathrm{op}}^2 + \|W\|_{\mathrm{op}}^2\right] \cdot \mathbb{E}_x[\|\mathscr{C}_x\|^2]. \tag{33}$$

Finally, denoting $\gamma_j = \hat{p}_\theta(y = j|x) - p(y = j|x)$ to ease notations, we notice that

$$\|\mathscr{C}_x\|^2 = \left\|\sum_{j=1}^{p}(\hat{p}_\theta(y = j|x) - p(y = j|x))E(j)\right\|^2$$
$$= \left\|\sum_{j=1}^{p}\gamma_j E(j)\right\|^2$$
$$= \sum_{l=1}^{d}\left(\sum_{j=1}^{p}\gamma_j E(j)_l\right)^2$$
$$\leq \sum_{l=1}^{d}\left(\sum_{j=1}^{p}|\gamma_j||E(j)_l|\right)^2$$
$$\leq \left(\sum_{l=1}^{d}\max_{j\in[p]}|E(j)_l|^2\right)\left(\sum_{j=1}^{p}|\gamma_j|\right)^2$$
$$\leq dB^2\left(\sum_{j=1}^{p}|\gamma_j|\right)^2. \qquad (\text{using } \|E\|_\infty \leq B)$$

Recalling that the total variation distance verifies for any probability vectors $p, q$

$$d_{\mathrm{TV}}(p, q) = \frac{1}{2}\|p - q\|_1 = \frac{1}{2}\sum_{\omega\in\Omega}|p(\omega) - q(\omega)|,$$

with $\|\cdot\|_1$ the $\ell_1$-norm. We notice that

$$\sum_{j=1}^{p}|\gamma_j| = \sum_{j=1}^{p}|\hat{p}_\theta(y = j|x) - p(y = j|x)| = \|\hat{p}_\theta(\cdot|x) - p(\cdot|x)\|_1 = 2 \cdot d_{\mathrm{TV}}(\hat{p}_\theta(\cdot|x), p(\cdot|x)).$$

Recalling that the total variation distance takes values in $[0, 1]$ and that $x \mapsto x^2 \le x \mapsto x$ on this interval, we know that $d_{\mathrm{TV}}(\hat{p}_\theta(\cdot|x), p(\cdot|x))^2 \le d_{\mathrm{TV}}(\hat{p}_\theta(\cdot|x), p(\cdot|x))$. It leads to

$$\mathbb{E}_x\big[\|\mathscr{C}_x\|^2\big] \le 4dB^2 \mathbb{E}_x[d_{\mathrm{TV}}(\hat{p}_\theta(\cdot|x), p(\cdot|x))].$$

Rearranging the terms in front of the expectation in (33) finally leads to

$$\|\nabla_\theta \mathcal{L}(\theta)\|^2 \le \hat{B} \cdot \mathbb{E}_x[d_{\mathrm{TV}}(\hat{p}_\theta(\cdot|x), p(\cdot|x))]$$

where $\quad \hat{B} \quad = \quad 4dB^2\bigg[\Big[2n\sqrt{d}\|V\|_{\mathrm{op}}(1 + \sqrt{\tfrac{2}{\pi}}\|W\|_{\mathrm{op}}^2\|U\|_{\mathrm{op}})\Big]^2 + d(1 + 2\|W\|_{\mathrm{op}}\|U\|_{\mathrm{op}})^2 + 4\|U\|_{\mathrm{op}}^2 + \|W\|_{\mathrm{op}}^2\bigg].$
Noticing that $\hat{B} = \mathcal{O}\Big(B^2\Big[\|V\|_{\mathrm{op}}\big[1 + \|U\|_{\mathrm{op}}\|W\|_{\mathrm{op}}^2\big]^2 + [1 + \|U\|_{\mathrm{op}}\|W\|_{\mathrm{op}}]^2 + \|U\|_{\mathrm{op}}^2 + \|W\|_{\mathrm{op}}^2\Big]\Big)$ and taking the square root leads to

$\|\nabla_\theta \mathcal{L}(\theta)\|$
$$= \mathcal{O}\Big(B\sqrt{\|V\|_{\mathrm{op}}\big[1 + \|U\|_{\mathrm{op}}\|W\|_{\mathrm{op}}^2\big]^2 + [1 + \|U\|_{\mathrm{op}}\|W\|_{\mathrm{op}}]^2 + \|U\|_{\mathrm{op}}^2 + \|W\|_{\mathrm{op}}^2} \cdot \sqrt{\mathbb{E}_x[d_{\mathrm{TV}}(\hat{p}_\theta(\cdot|x), p(\cdot|x))]}\Big),$$

which concludes the proof. $\qquad\square$

