# OpenReview forum: "A Mechanistic Study of Transformers Training Dynamics"
_TMLR — Withdrawn by Authors_

### Review · Reviewer_3efb · 2026-03-24

**Summary Of Contributions:**

This work investigates the training dynamics for a Transformer model by mechanically analyzing the components during training. In this study, a synthetic task, sparse modulo addition, is employed as a benchmark dataset to analyze the behavior of Transformer training, which basically performs prefix sums with modulo so that a model should be aware of the prefix positions and the value in each position. A single layer Transformer model is employed with a pre-normalization configuration, which basically ask the model to summarizes the inputs to yield a single vector representation, re-interpreted after an MLP, followed by a single head prediction. Experiments show that the training seems to be split into two phases. The spikes during training also indicates the two root causes, one from RMS normalization and the other from MLP.

Strengths
- A synthetic benchmark is well motivated to investigate the training dynamics of a model by solving a task of identifying the prefix positions and their associated values to find an answer using modulo.
- Experiments show interesting observations of training in two phases and the causes of loss spikes, which justifies the analyses and experiments in prior studies.

Weaknesses
- It is not clear whether findings by the toy single-layer Transformer model with $d=2$ could be generalized to the more complex models with several layers.
- The investigated model is an encoder model, not an auto-regressive decoder model employed in conventional language models. Thus, it is not clear whether the analyses hold for such a decoder model.
- No explanation exists for the optimizer and learning rate scheduling, and thus, the interaction of the training progress with the learning rate is not clear.
- The variances of training runs are not discussed.

**Audience:**

Yes

**Audience Explanation:**

The findings of this work could have an impact for the future studies in understanding the Transformer model.

**Claims And Evidence:**

No

**Claims Explanation:**

The training dynamics are usually influenced by the choice of optimizers and learning rate curves, not only the model settings. Given that the optimizer and learning rate schedule employed in the experiments are not documented in this paper, it is not clear whether we can draw any conclusions.

**Requested Changes:**

- I'd like to see an impact of the choice of optimizer and learning rate schedule to see how that will impact the training dynamics.
- It sounds like the analysis is based on a single run using a particular random seed setting, but this work needs to show the variances of training run to see whether the findings are consistent across runs.
- An additional discussion is necessary to interpret the findings on a single-layer encoder-style Transformer model, for a more broader decoder-style model and multiple-layer settings.

---

### Review · Reviewer_BYC3 · 2026-03-31

**Summary Of Contributions:**

This paper studies the training dynamics of a simple transformer model on a synthetic task of Sparse Modular Addition. It visualizes patterns of different components of the transfer model during the training process, including attention, Token Embeddings, Laws,Gradient norms, Activation Sparsity, and the MLP classifier decision regions. Experiments on a single-layer attention transforming block, With lower dimensionality Hidden states such as two dimensions, Discover different patterns of how the model dynamically evolves during training on this particular task. These findings includeSensitivity of attention map to initialization of parameters, Sparsity of neurons, a two-stage gloss drop-in learning, And the correlation of lost spikes of gradient norms. The hope is to use intuition from the small-scale synthetic task training to learn patterns to get practical language model development.

**Strengths:**

The paper is mostly well written, presenting a clear view from the visualization analysis.

There are interesting findings reviewed by the small-scale experiments, such as the attention sensitivity to the initialization, the double descent of the loss, and the explanation behind that.

The paper also tries to run different experiments in the appendix to show results beyond the two-dimensional case.

**Weaknesses:**

Some of the descriptions of the model may not be fully accurate. For example, The introduction of cross-attention Used as a simple language model It is not very clear. Specifically, for a decoder language model, the mechanism is implemented as masked self-attention. Well, this is only a technicality, Clarity of the description is important in understanding the details.

One big concern of this paper (and generally similar lines of work) is that the small-scale experiments do not necessarily translate to large-scale language model training dynamics. The Transformer model is a very much simplified version without deep layers, The task is a very simple rule-based sparse modular addition task, And the model dimension is restricted to two for the majority of the partitions For the ease of visualization. At the end of the introduction, The paper claims that “study training stability in small scale to gain intuition for larger-scale settings”, but there is no real evidence presented that any study and results shown here can be effectively translated to large-scale language models that we use today.

That being said, the paper does a good job in citing relevant references and confirming similar findings. However, considering some of the findings were already discussed in prior literature, it also remains a question how many novel takeaways from this paper can be transferred to new setups. For example, Towards the end of section 4, “Implication for large language models”, the paper suggests that “reducing gradient norms … can help stabilize the training”, but gradient norm clipping is already a widely used technique in most of the model training, even back in the days when we train LSTM based small language models before Transformers. The indications to large language models are really not very clear.

**Additional Comments:**

Attention pattern depends on the initialization, shown in Figure 2: why at the beginning there is a pattern of low attention on some positions consistently? Is this from random initialization?

**Audience:**

Yes

**Audience Explanation:**

Although the indication to large and real language models is unclear, the paper does a good job of rigorously conducting experiments of a simple transformer training on a synthetic task. The visualizations are thoroughly discussed, with good indications as a takeaway in this particular example. It could be valuable to certain researchers who are interested in this kind of small-scale white box analysis.

**Claims And Evidence:**

No

**Claims Explanation:**

Some of this is discussed in the weaknesses above. For example, at the end of the introduction, the paper claims that “study training stability in small scale to gain intuition for larger-scale settings”, but there is no real evidence presented that any study and results shown here can be effectively translated to large-scale language models that we use today.

For another example, the Theoretical Justification in Section 3 does not fully establish the claim that “In the previous section, we showed that these pathways could be effectively learned by gradient descent.” from Section 4. Section 3 just simply provides a possible solution that can learn the task as one of the optimal, But it does not establish any effectiveness of the gradient descent to converge to this particular solution framed as “clustering heads.” (By the way, I think a named “clustering head” is a bit confusing, since in LLM interpretability, there are many works that use "heads" to refer to specific attention heads that show a particular property. Whereas in this paper, I believe there's no multi-head attention.)

Furthermore, the analysis is based on visualizations, which is not a new contribution In both methodology and analysis approach. The use of the word "sandbox" is also a bit overclaimed. There are no quantitative measurements in the paper to support The claims; all conclusions and results are purely observational based on qualitative results.

**Requested Changes:**

The paper could benefit from more accurate use of the terminology around language model research, such as having a clear definition/use of terms like “cross-attention”, “visual sandbox”, “clustering heads”.

Not sure how this translates to real language models, and at a larger scale; there is no evidence that this would transfer.

The paper could also have a better separation of what was already known vs. what was discovered as new, such as the findings of gradient norm and other phenomena.

For other suggestions for revision, please see the weaknesses.

---

### Review · Reviewer_Vf3v · 2026-04-06

**Summary Of Contributions:**

This work studies the training dynamics of a single-block transformer on the sparse modular addition task. The authors propose a novel concept called "clustering heads", which they call attention circuits that capture permutation and suffix invariances. They design a 2D visual sandbox to track token embeddings during training, which reveals a two-stage learning process (embedding clustering followed by MLP fitting) and attributing loss spikes to high curvature in normalization and feedforward layers. Additional experiments examine activation sparsity, circuit transferability, and the role of initialization.

**Additional Comments:**

Language and presentation issues:

- "This showcases the benefits of our setup in studying, at a small scale, the training behavior of bigger models." (p.2). This is asserted without evidence at that point in the paper; the connection to larger models is not established.

"waiting for a clear signal to escape it" (p.7). Anthropomorphizing the optimization process without formal justification.

- Section 4.4: "One interesting aspect of our sandbox is that it generates loss spikes that we can study quite precisely.". Self-congratulatory framing; better to simply present the findings.

- "Parasitic influence" (p.1): Used to describe potential variables the authors want to exclude. "Confounding factors" would be more appropriate.


Typos and minor issues:
- p.5: "in the plan" → "in the plane" (Section 4.1, referring to d=2 embedding)
- p.7: "pics" → "peaks" (Figure 6 caption, "the three pics correspond to the loss drops")
- p.8: "pics" → "peaks" (same issue repeated)
- p.2: The notation [n] = {1,...,n} is introduced but n is also used for vocabulary size. Later p is introduced for vocabulary size in some equations but n is used in the main text. In general, the notation for vocabulary is at times inconsistent and needs to be improved.
- p.3: "merge the query and value into a vector q". This is confusing since q typically denotes the query, not a merged query-value vector. The sentence "We omit the key matrix" also needs more justification.

**Audience:**

Yes

**Audience Explanation:**

Yes, with reservations. The general direction of using controlled settings and mechanistic interpretability to understand transformer training is of vry broad interest. The visual sandbox idea is itself very appealing as an exploratory tool. However, the current work does not deliver sufficiently validated or novel insights to meet TMLR's bar for contributions that "would be of interest to the TMLR audience." The two-stage learning observation echoes well-documented findings in the in-context learning literature (Olsson et al., 2022; Edelman et al., 2024), and the loss spike analysis, does not go beyond what is already understood about normalization-induced instabilities.

**Broader Impact Concerns:**

No concerns regarding ethics or broader impact.

**Claims And Evidence:**

No

**Claims Explanation:**

Issue 1: Experimental setup is too narrow to justify the claims.
The core experiments use a single transformer block with d=2 embedding dimensions, vocabulary size n∈{2,3}, and sequence length L=12.The gap between this setup and practical transformers is so large that the claimed connections to LLM pretraining dynamics (loss spikes, stage-wise learning) are not justified. The paper often draws parallels to large-scale training phenomena but provides no intermediate-scale validation justify this strong claim.


Issue 2: Alternative methods to clustering heads were not explired.
The authors acknowledge that clustering heads are not the only possible circuit, and Figures 11 through 13 show that the practically learned circuits deviate substantially from the idealized version (faulty attention compensated by value matrices, partially learned invariants). Yet the theoretical analysis (Propositions 1 and 3) only establishes that the idealized clustering head is a stationary point. The paper thus cannot claim that gradient descent preferentially finds these circuits. The gap between the idealized theory and the fairly messy empirical circuits undermines the explanatory power of the framework.

Issue 3: Theoretical contribution is weak.
While the paper provides a rigorous derivation of the gradients for a specific transformer setup, the theoretical insights derived from those gradients (Propositions 1 & 2) are largely marginal. Proposition 1 shows that a perfect classifier is a stationary point of the cross-entropy loss. This is almost tautological: any parameter configuration achieving zero loss satisfies ∇L(θ)=0 because the residuals p̂−p vanish, making all gradient terms zero. The proposition does not provide convergence guarantees, which the authors also acknowledge.  Proposition 2's gradient upper bound, while also technically sound, offers limited insight beyond the observation that gradients shrink as the model improves. The authors' interpretation is that in early iterations, "poor behavior" leads to "weak control" (large gradients), while improvement leads to "strengthened control" (smaller gradients). This is a standard property of most smooth, non-convex loss functions used in deep learning and does not offer a new "mechanistic" insight into how the transformer architecture specifically interacts with this bound

Issue 4: Insufficient investigation of positional embeddings.
For the SMA task, correctly distinguishing spurious from non-spurious positions is fundamental. Yet the paper does not investigate how different positional embedding schemes (learned vs. sinusoidal, absolute vs. relative) affect learned circuits. The visual sandbox shows position embeddings collapsing, but there is no systematic ablation studying whether this collapse is robust or fragile, and how it depends on the positional encoding type.

Issue 5: Claims regarding loss spike are too strong.
Section 4.4 draws connections between the observed loss spikes and those in large-scale pretraining, citing QK-Norm and MuonClip as heuristics. However, the mechanisms causing loss spikes in a single-block d=2 transformer on a synthetic task may differ fundamentally from those in billion-parameter models trained on natural language. The paper suggests smoothing normalization layers as a mitigation strategy but it is untested.

**Requested Changes:**

**Critical**

Change 1: Strengthen motivation and connection to prior work.
The choice of SMA as the task and clustering heads as the circuit of interest needs substantially better justification. The paper should clearly articulate what SMA captures that prior synthetic tasks (modular addition without sparsity, sparse parity, in-context learning setups) do not, and why the findings are expected to transfer. The related work section should engage more deeply with concurrent and recent mechanistic interpretability work on circuit formation during training (not just post-hoc circuit discovery).

Change 2: Characterize alternative circuits
The paper should move beyond showing that clustering heads are a stationary point and investigate what other solutions the transformer finds. Figures 11–13 show diverse circuits which deserve systematic categorization and analysis of when each type emerges, going further than anecdotal presentation.

Change 3: Validate beyond d=2.
The clustering head mechanism should be verified at higher dimensions using quantitative metrics (e.g., measuring invariance violation, cluster quality scores) rather than relying solely on 2D visualization. The current d>2 experiments only track aggregate statistics (loss, gradient norms, sparsity).

Change 4: Position embedding ablations.
Given that the task fundamentally depends on distinguishing positions 1–k from k+1–L, the paper must study how positional embedding type, initialization scale, and dimensionality affect the learned circuits and training dynamics.

Change 5: Weaken LLM transferability claims.
Either provide concrete evidence connecting these findings to larger-scale phenomena (e.g., replicate on a multi-layer transformer, or show the suggested normalization smoothing actually helps) or significantly tone down the claims about implications for LLM training.


**Suggested**

- Multi-head and multi-layer extensions.
Even a modest extension to 2 layers or 2 heads would significantly strengthen the claims about practical relevance.

- Quantitative phase transition analysis.
The two-stage learning is only identified visually. Formal metrics (e.g., representational similarity analysis or probing classifiers for invariance) would make the claims more rigorous.
- Compare with grokking literature. The SMA task and the observed training dynamics (slow initial phase, sudden generalization) are reminiscent of grokking (Power et al., 2022; Nanda et al., 2023, which are cited but not deeply engaged with). A clearer positioning relative to this body of work would strengthen the contribution.


Literature that needs to be more adequately addressed:

**Grokking literature**
- Liu et al. (2022), "Towards understanding grokking: An effective theory of representation learning"
- Kumar et al. (2024), "Grokking as the transition from lazy to rich training dynamics"
- Morwani et al. (2024), "Feature emergence via margin maximization: case studies in algebraic tasks"

**Circuit discovery**
- Conmy et al. (2023), "Towards Automated Circuit Discovery for Mechanistic Interpretability"

**LLM training and loss spikes**
- Takase et al. (2024), "Spike No More: Stabilizing the Pre-training of Large Language Models"
-  more thorough engangement with "Edge of stability" works from Cohen et al. 2022 and subsequent literature.

---

### Note · Authors · 2026-05-09

**Comment:**

Dear reviewers,

We would like to thank the reviewers for taking the time to carefully read our submission and providing helpful feedback.

Several suggestions were provided to improve the support of our claims. We believe that a thorough treatment is needed to adequately address them. For this reason, we prefer to withdraw the paper to save reviewers and AC time. We will take the time to integrate our new findings for a more comprehensive study that will be subject to a future submission.

We thank the reviewers again for their comments that were useful in improving our work and that gave us many interesting directions for future work.

Best regards,

The authors.

**Withdrawal Confirmation:**

I have read and agree with the venue's withdrawal policy on behalf of myself and my co-authors.